# Minibatch selection for Language Models
# via Partition Matroid Constrained Gradient Matching

**Prayas Agrawal** [* 1]   **Prateek Chanda** [* 2]   **Ishita Khatri** [3]   **Ganesh Ramakrishnan** [2 3]   **Bamdev Mishra** [4]   **Pratik Jawanpuria** [3]

## Abstract

Training large language models (LLMs) on heterogeneous data requires selecting minibatches that balance convergence speed with coverage across domains. Existing methods either select samples independently within each domain or rely on computationally expensive proxy models to learn continuous domain weights. We propose PARTITIONSEL, a cross-domain minibatch selection approach that maximizes a validation-guided gradient-matching utility under per-domain budgets encoded as a partition-matroid constraint. By coupling the per-domain budgets through a single utility, PARTITIONSEL is designed to reduce redundancy in selections across domains. The proposed objective is weakly submodular and admits an orthogonal matching pursuit algorithm with provable approximation guarantees. Empirically, we evaluate PARTITIONSEL for minibatch selection during the fine-tuning of QWEN2.5 and LLAMA-3 on METAMATHQA and MOL-INSTRUCTIONS. PARTITIONSEL achieves robust gains over per-domain and domain-agnostic baselines on both benchmarks. It also reduces the number of conflicting gradient pairs within each batch, indicating that the cross-domain coupling translates into more compatible training updates. Code is available here.

---

[*]Equal contribution  [1]Microsoft Research India [2]Department of Computer Science and Engineering, Indian Institute of Technology Bombay [3]Centre for Machine Intelligence and Data Science, Indian Institute of Technology Bombay [4]Microsoft India. Correspondence to: Prayas Agrawal <agrawalprayas9@gmail.com>, Prateek Chanda <prateekch@cse.iitb.ac.in>, Pratik Jawanpuria <pratik.jawanpuria@iitb.ac.in>.

*Proceedings of the $43^{rd}$ International Conference on Machine Learning*, Seoul, South Korea. PMLR 306, 2026. Copyright 2026 by the author(s).

## 1. Introduction

Training large language models (LLMs) with larger mini-batches is known to improve convergence rates and can lead to superior performance. However, large mini-batches incur substantial GPU memory overhead. Recent works propose selectively training on targeted samples within each mini-batch (Wang et al., 2024; Nguyen et al., 2025), thereby significantly reducing memory consumption. The selection of such samples is typically performed using influence functions (Fahrbach et al., 2019; Pruthi et al., 2020; Gasteiger et al., 2022), gradient-matching techniques (Mirzasoleiman et al., 2020a; Killamsetty et al., 2021b;a), or distribution based matching approaches (Kim et al., 2016; Gurumoorthy et al., 2019; 2021; Liu et al., 2024b; Chanda et al., 2026).

The minibatch selection problem becomes more challenging when the data includes multiple heterogeneous domains. Several works emphasize on identifying optimal domain representations during training (Tishby & Zaslavsky, 2015; Maurer et al., 2016; Namkoong & Duchi, 2016). Consequently, recent works (Xie et al., 2023a; Liu et al., 2025) aim at optimizing the domain representation mixture via proxy models which are trained alongside the main LLM. However, training proxy models involve computational and practical concerns as one needs to ensure that the proxy model faithfully mirrors the optimization trajectory or emergent behaviors of the main LLM.

In this work, we take a discrete, combinatorial view of domain-aware minibatch construction that sidesteps the need for any auxiliary proxy model. We encode per-domain capacities directly as a partition-matroid constraint on the minibatch and select samples jointly under this constraint by maximizing a common utility function across domains. Overall, the samples are rewarded not only for aligning with the signals within their own domain, but also for being non-redundant with samples chosen from other domains. The resulting objective is weakly submodular and admits an efficient orthogonal matching pursuit algorithm with provable approximation guarantees – effectively inducing an implicit domain mixture at the batch level without the computational overhead of training a proxy alongside the LLM. In summary, our contributions are as follows:

- **A unified, domain-coupled formulation.** We cast cross-domain minibatch selection as the maximization of a single, validation-guided gradient-matching utility subject to per-domain budgets encoded as a partition-matroid constraint. The proposed weighted-prototype objective, PARTITIONSEL, recovers the GREATS selection rule (Wang et al., 2024) as the binary special case, while, unlike GREATS ,being provably monotone and weakly submodular.

- **A discrete bridge to continuous data-mixture methods.** PARTITIONSEL offers a principled connection between discrete subset selection and continuous reweighting strategies such as DoReMi (Xie et al., 2023a), by inducing a discrete, sample-level domain mixture at each training step.

- **Empirical gains and gradient-level evidence.** Across QWEN2.5 and LLAMA-3 fine-tuned on META-MATHQA and MOL-INSTRUCTIONS, PARTITIONSEL based minibatch selection consistently improves upon strong baselines including GREATS (Wang et al., 2024), COLM (Nguyen et al., 2025), and DoReMi (Xie et al., 2023a). PARTITIONSEL also reduces the number of conflicting gradient pairs within each batch, indicating that the proposed cross-domain coupling translates into more compatible training updates.

## 1.1. Related Works

**Coresets and subset selection for language models.** Coreset methods select small training subsets whose aggregate gradient approximates that of the larger dataset. CRAIG (Mirzasoleiman et al., 2020b) and GradMatch (Killamsetty et al., 2021a) formalize this as gradient-matching, while GLISTER (Killamsetty et al., 2021b) adds a bilevel validation objective. Recent works extend these ideas to LLMs at the mini-batch level: CoLM (Nguyen et al., 2025) performs memory-efficient coreset selection in the Adam-normalized gradient space for fine-tuning on heterogeneous mixtures while UniProT (Chanda et al., 2026) selects uniformly weighted mini-batch selection for improving minority class representation. Our work shares this gradient-matching philosophy but *couples* per-domain selections through a single validation-guided utility under partition-matroid constraints.

**Validation-guided and influence-based selection.** A complementary line of work scores examples by their estimated influence on a held-out signal using influence functions or gradient-based approximations (Koh & Liang, 2017; Pruthi et al., 2020). LESS (Xia et al., 2024b) computes low-rank gradient features for targeted instruction tuning, and GREATS (Wang et al., 2024) proposes an efficient gradient-matching selection rule at the mini-batch level. Quality-scoring methods such as Ask-LLM and QuRating (Sachdeva et al., 2026; Wettig et al., 2024) use model-derived signals

to filter pretraining data, while DSIR (Xie et al., 2023b) performs importance resampling against a target distribution. PARTITIONSEL similarly leverages validation guidance, but performs *joint* selection across domains under explicit per-domain capacity constraints, which we show reduces redundancy and gradient conflicts within each batch.

**Continuous data-mixture and reweighting methods.** Rather than selecting discrete subsets, mixture methods learn continuous domain weights to reweight the training distribution. DoReMi (Xie et al., 2023a) optimizes the mixture via a group-DRO proxy, RegMix (Liu et al., 2025) fits a regression surrogate over small-scale runs, and CLIMB (Diao et al., 2025) iteratively bootstraps clusters and weights. MATES (Yu et al., 2024b) learns a model-aware data selector and Sheared-LLaMA (Xia et al., 2024a) couples data selection with (model) structured pruning. These approaches build on group-robust and distributionally robust optimization (Sagawa et al., 2019) but typically require auxiliary proxy models or extra optimization loops. Our batch-level constrained selection can be viewed as a discrete alternative that induces an *implicit* mixture without an auxiliary proxy.

## 2. Background

**Notations**. The $n$-dimensional non-negative orthant is denoted by $\mathbb{R}_+^n$. Sets are denoted by calligraphic notation (e.g. $\mathcal{S}$). For any set $\mathcal{A}$, the corresponding index set is denoted by $\overline{\mathcal{A}}$. Vectors are typeset in lowercase boldface (e.g., $\mathbf{x}$), and matrices are typeset in uppercase boldface (e.g., $\mathbf{X}$). Sub-vectors and sub-matrices indexed by index sets are denoted using bracketed subscripts (e.g., $\mathbf{x}_{[\overline{\mathcal{I}}]}$, $\mathbf{X}_{[\overline{\mathcal{I}}],[\overline{\mathcal{J}}]}$), where $\overline{\mathcal{I}}$ and $\overline{\mathcal{J}}$ are the index sets. The probability simplex set is denoted by $\Delta^n = \{\mathbf{x} \in \mathbb{R}_+^n : \sum_{i=1}^n x_i = 1\}$. The operator (spectral) norm of a matrix $\mathbf{A}$ is $\|\mathbf{A}\|_{\mathrm{op}} := \sup_{\|\mathbf{x}\|_2=1} \|\mathbf{A}\mathbf{x}\|_2$. We let $\uplus$ denote the disjoint union of sets and $\mathbb{I}[\cdot]$ the indicator function. For a vector $\boldsymbol{x} \geq \mathbf{0}$, $\mathrm{supp}(\boldsymbol{x}) = \{j : \boldsymbol{x}_j > 0\}$ denotes the *support* of $\boldsymbol{x}$.

**Submodularity and submodularity ratio**. Let $f : 2^{\mathcal{V}} \to \mathbb{R}$ be a set function over a ground set $\mathcal{V}$. For any subset $A \subseteq \mathcal{V}$ and element $e \in \mathcal{V} \setminus A$, the marginal gain is defined as $\mathbf{\Delta} f(e|A) = f(A \cup \{e\}) - f(A)$. The function $f$ is submodular if it exhibits diminishing returns: for all $A \subseteq B \subseteq \mathcal{V}$ and $e \in \mathcal{V} \setminus B$, $\mathbf{\Delta} f(e|A) \geq \mathbf{\Delta} f(e|B)$. The submodularity ratio $\gamma$ captures how close a non-submodular function is to being submodular: $f$ is said to be $\gamma$-weakly submodular for some $\gamma \in (0, 1]$ if it satisfies: $\sum_{e \in B} \mathbf{\Delta} f(e|A) \geq \gamma \mathbf{\Delta} f(B|A)$ for all disjoint sets $A, B \subseteq \mathcal{V}$, where $\mathbf{\Delta} f(B \mid A) = f(A \cup B) - f(A)$. For submodular functions, $\gamma = 1$. To maximize a monotonic $\gamma$-weakly submodular function under cardinality constraint, the classical greedy algorithm (based on marginal gains) achieves $(1 - e^{-\gamma})$ approximation of the optimal value (Nemhauser et al., 1978; Das & Kempe, 2018).

**RSC and RSM**. A differentiable function $l : \mathbb{R}^d \to \mathbb{R}$ is *restricted strong concave* (RSC) with parameter $c_\Omega$ and *restricted smooth* (RSM) with parameter $C_\Omega$ on a domain $\Omega \subset \mathbb{R}^d \times \mathbb{R}^d$ if $\forall (\boldsymbol{x}, \mathbf{y}) \in \Omega$,

$$l(\boldsymbol{x}) - l(\mathbf{y}) + \langle \nabla l(\boldsymbol{x}), \mathbf{y} - \boldsymbol{x} \rangle \in [\frac{c_\Omega}{2} \|\mathbf{y} - \boldsymbol{x}\|_2^2, \frac{C_\Omega}{2} \|\mathbf{y} - \boldsymbol{x}\|_2^2].$$

RSC and RSM have been studied in the context of weakly submodular functions (Elenberg et al., 2018; Gurumoorthy et al., 2019). Let $\Omega_\kappa := \{(\mathbf{x}, \mathbf{y}) : \|\mathbf{x}\|_0 \leq \kappa, \|\mathbf{y}\|_0 \leq \kappa, \mathbf{x}, \mathbf{y} \geq 0\}$, and let $c_\kappa$ and $C_\kappa$ denote the RSC and RSM parameters of $l$ on $\Omega_\kappa$. Since $\Omega_k \subseteq \Omega_\kappa$ whenever $k \leq \kappa$, it is easily verified that $c_k \geq c_\kappa$ and $C_k \leq C_\kappa$. Finally, define $\widetilde{\Omega} := \{(\mathbf{x}, \mathbf{y}) : \|\mathbf{x} - \mathbf{y}\|_0 \leq 1\}$, with associated smoothness parameter $\widetilde{C}_1$.

**Partition matroid**. A partition matroid $\mathcal{M} = (\mathcal{V}, \mathcal{I})$ is defined by a partition of the ground set $\mathcal{V}$ into disjoint sets $\mathcal{V}_1, \ldots, \mathcal{V}_m$ (i.e., $\mathcal{V} = \uplus_{i=1}^m \mathcal{V}_i$) with capacities $k_1, k_2, \ldots, k_m$, respectively. Here $\mathcal{I}$ is the set of all *independent* subsets of $\mathcal{V}$, defined as $\mathcal{I} = \{\mathcal{S} \subseteq \mathcal{V} \mid |\mathcal{S} \cap \mathcal{V}_i| \leq k_i \forall i = 1, \ldots, m\}$.

**Conflicting gradients**. We define $\phi_{ij}$ as the angle between two sample gradients $\mathbf{g}_i$ and $\mathbf{g}_j$. The gradients are *conflicting* if $\cos \phi_{ij} < 0 (\equiv \langle \mathbf{g}_i, \mathbf{g}_j \rangle < 0)$. In Section 7.4, we use this notion of gradient conflict to analyze how samples from different domains interfere with one another during training.

# 3. Problem Formulation

We consider the standard setting for training a language model. Let $\boldsymbol{D}_{\text{tr}}$ denote the training dataset and $\ell : \mathcal{Z} \times \boldsymbol{\Theta} \to \mathbb{R}$ be an instance-wise loss function, where $\mathcal{Z} = \mathcal{X} \times \mathcal{Y}$ is the instance space and $\boldsymbol{\Theta}$ is the model parameter space. We seek to minimize the expected loss via iterative stochastic optimization (e.g., Adam). At each step $t \in [T]$, the model receives a mini-batch $\mathcal{B}_t \subseteq \boldsymbol{D}_{\text{tr}}$, and the parameters $\boldsymbol{\theta}_t$ are updated using the cumulative batch loss $\boldsymbol{L}_t(\boldsymbol{\theta}_t) = \sum_{\mathbf{z} \in \mathcal{B}_t} \ell(\mathbf{z}; \boldsymbol{\theta}_t)$. To guide the learning process, we assume access to a small validation set $\mathcal{D}_{\text{val}}$ (Wang et al., 2024; Xia et al., 2024b; Xie et al., 2023a), from which a validation mini-batch $\mathcal{B}_t^{\text{val}} \subseteq \mathcal{D}_{\text{val}}$ is sampled at each step.

**Domain heterogeneity and mixture strategies**. We consider the setting where the data distribution is heterogeneous, i.e., the training dataset spans $C$ distinct domains. Accordingly, $\boldsymbol{D}_{\text{tr}}$ decomposes as $\boldsymbol{D}_{\text{tr}} = \uplus_{c=1}^C \mathcal{V}_c$, where $\mathcal{V}_c$ denotes the samples belonging to the $c$-th domain. Recent domain-aware mixture strategies typically learn an optimal continuous weight vector $\mathbf{w} \in \Delta^C$ over the domains. However, estimating these weights requires computationally expensive techniques such as training auxiliary proxy models to predict downstream performance (Liu et al., 2025; Diao et al., 2025) or employing group distributionally robust optimization (Xie et al., 2023a; Sagawa et al., 2019).

**Domain-aware batch selection**. To circumvent the computational costs of the above techniques, we reformulate the problem as domain-aware mini-batch selection. Instead of learning global domain weights, we optimize the data mixture *locally*, at the batch level, by selecting a subset of samples that maximizes a utility function subject to domain constraints. For a given training batch $\mathcal{B}_t$ at step $t$, we consider the disjoint union of samples across all $C$ domains, $\mathcal{B}_t = \uplus_{c=1}^C \mathcal{B}_t^c$, where $\mathcal{B}_t^c = \mathcal{B}_t \cap \mathcal{V}_c$. Our objective is to select a subset $\mathcal{S}_t \subseteq \mathcal{B}_t$ that maximizes a time-dependent utility $\mathcal{U}_t(\mathcal{S})$ while respecting per-domain cardinality constraints $\{\kappa_c^t\}_{c=1}^C$. We define the selection problem as:

$$\max_{\mathcal{S} \subseteq \mathcal{B}_t} \mathcal{U}_t(\mathcal{S}) \quad \text{s.t.} \quad |\mathcal{S} \cap \mathcal{B}_t^c| \leq \kappa_c^t \, \forall c, \sum_{c=1}^C \kappa_c^t = \kappa \quad (1)$$

where $\kappa$ is the per-step selection budget. The constraint in (1) is precisely a partition-matroid constraint with capacities $\{\kappa_c^t\}$. This formulation lets us enforce domain-specific budgets (and effectively inducing a data mixture) without proxy models or continuous weight optimization.

## 3.1. Validation-Guided Gradient Matching

The utility $\mathcal{U}_t(\mathcal{S})$ quantifies the marginal value of training samples by balancing alignment with the validation signal (to ensure relevance) against intra-subset diversity (to limit redundancy and gradient conflicts). To instantiate it, we adopt a validation-guided approach: we select a training subset $\mathcal{S} \subseteq \mathcal{B}_t$ such that the model update derived from $\mathcal{S}$ maximally reduces the loss on the validation batch $\mathcal{B}_t^{\text{val}}$. Let $\boldsymbol{\theta}_t$ be the current model parameters. Following (Koh & Liang, 2017; Killamsetty et al., 2021b; Xia et al., 2024b; Wang et al., 2024), we define the utility of a set $\mathcal{S} \subseteq \mathcal{B}_t$ as the expected reduction in validation loss after a stochastic gradient-descent step with learning rate $\eta_t$. For a validation instance $\mathbf{z}_{\text{val}}$, the expected reduction in validation loss is

$$\mathcal{U}_t(\mathcal{S}) := \ell(\mathbf{z}_{\text{val}}; \boldsymbol{\theta}_t) - \ell\Big(\mathbf{z}_{\text{val}}; \tilde{\boldsymbol{\theta}}_{t+1}(\mathcal{S})\Big),$$

where $\tilde{\boldsymbol{\theta}}_{t+1}(\mathcal{S}) = \boldsymbol{\theta}_t - \eta_t \sum_{\mathbf{z} \in \mathcal{S}} \boldsymbol{g}_{\boldsymbol{\theta}_t}(\mathbf{z})$ denotes the parameters updated using the subset $\mathcal{S}$, and $\boldsymbol{g}_{\boldsymbol{\theta}_t}(\mathbf{z}) := \nabla_{\boldsymbol{\theta}} \ell(\mathbf{z}; \boldsymbol{\theta})|_{\boldsymbol{\theta} = \boldsymbol{\theta}_t}$ is the gradient of $\ell(\mathbf{z}; \boldsymbol{\theta})$ at $\boldsymbol{\theta}_t$.

Given the partially constructed selection $\mathcal{S} := \{\mathbf{z}_1, \mathbf{z}_2, \cdots, \mathbf{z}_{i-1}\}$ at step $t$, we compute the marginal gain of the utility as follows:

$$\Delta \mathcal{U}_t(\mathbf{z}_i \mid \mathcal{S}) = \ell(\mathbf{z}_{\text{val}}; \tilde{\boldsymbol{\theta}}_{t+1}(\mathcal{S})) - \ell(\mathbf{z}_{\text{val}}; \tilde{\boldsymbol{\theta}}_{t+1}(\mathcal{S} \cup \{\mathbf{z}_i\}))$$
$$\approx \eta_t \left\langle \boldsymbol{g}_{\boldsymbol{\theta}_t}(\mathbf{z}_i), \boldsymbol{g}_{\boldsymbol{\theta}_t}(\mathbf{z}_{\text{val}}) \right\rangle$$
$$- \eta_t^2 \left\langle \boldsymbol{g}_{\boldsymbol{\theta}_t}(\mathbf{z}_i), \boldsymbol{H}_{\mathbf{z}_{\text{val}}}(\boldsymbol{\theta}_t)(\sum_{\mathbf{z} \in \mathcal{S}} \boldsymbol{g}_{\boldsymbol{\theta}_t}(\mathbf{z})) \right\rangle.$$
$$(2)$$

Equation (2) follows from two successive first-order Taylor expansions (Wang et al., 2024): a first-order expansion of the loss difference, together with the update relation

$\tilde{\boldsymbol{\theta}}_{t+1}(\mathcal{S} \cup \{\mathbf{z}_i\}) = \tilde{\boldsymbol{\theta}}_{t+1}(\mathcal{S}) - \eta_t \, \boldsymbol{g}_{\boldsymbol{\theta}_t}(\mathbf{z}_i)$, followed by a first-order expansion of the validation gradient $\nabla_{\boldsymbol{\theta}} \ell(\mathbf{z}_{\mathsf{val}}; \cdot)$ about $\boldsymbol{\theta}_t$. Here $\boldsymbol{H}_{\mathbf{z}_{\mathsf{val}}}(\boldsymbol{\theta}_t) := \nabla_{\boldsymbol{\theta}}^2 \ell(\mathbf{z}_{\mathsf{val}}; \boldsymbol{\theta})|_{\boldsymbol{\theta}=\boldsymbol{\theta}_t}$ denotes the validation-loss Hessian at $\boldsymbol{\theta}_t$.

The first term in (2) is the importance score of $\mathbf{z}_i$ with respect to the validation point $\mathbf{z}_{\mathsf{val}}$. It captures how effectively the gradient of the training instance $\mathbf{z}_i$ reduces the validation loss. The second term is the Hessian-weighted similarity of the candidate $\mathbf{z}_i$ to the training instances in $\mathcal{S}$. It is subtracted and hence promotes diversity (Wang et al., 2024).

### 3.2. GREATS (Wang et al., 2024)

Wang et al. (2024) proposed the GREATS algorithm, which approximates the vanilla greedy algorithm by iteratively updating the set as $\mathcal{S} \leftarrow \mathcal{S} \cup \{\mathbf{z}^*\}$ (while respecting the cardinality constraint $|\mathcal{S}| \leq \kappa$), where

$$\mathbf{z}^* := \underset{\mathbf{z} \in \mathcal{B}_t \setminus \mathcal{S}}{\arg\max} \quad \langle \boldsymbol{g}_{\boldsymbol{\theta}_t}(\mathbf{z}), \boldsymbol{g}_{\boldsymbol{\theta}_t}(\mathbf{z}_{\mathsf{val}}) \rangle \\ - \eta_t \left\langle \boldsymbol{g}_{\boldsymbol{\theta}_t}(\mathbf{z}), \sum_{\mathbf{z}_j \in \mathcal{S}} \boldsymbol{g}_{\boldsymbol{\theta}_t}(\mathbf{z}_j) \right\rangle. \quad (3)$$

Since the Hessian-gradient product is expensive, Wang et al. (2024) approximate $\boldsymbol{H} \approx \mathbf{I}$ in (2) to obtain (3).

It should be noted that, while (3) appears to be an instance of greedy selection that maximizes a specific marginal-gain function, Wang et al. (2024) observed (on the review-discussion forum) that it is not an instance of submodular maximization. They conjectured that the GREATS objective may "potentially fall into the category of weakly submodular functions." As discussed in Section 2, (weak) submodularity is a desirable property of set functions, since it yields an approximation guarantee for the vanilla greedy algorithm.

We also note that the marginal gain for an element $\mathbf{z} \in \mathcal{B}_t \setminus \mathcal{S}$ in (3) can easily be negative. A simple scenario is when a candidate's gradient is near-orthogonal to the validation gradient ($\langle \boldsymbol{g}_{\boldsymbol{\theta}_t}(\mathbf{z}), \boldsymbol{g}_{\boldsymbol{\theta}_t}(\mathbf{z}_{\mathsf{val}}) \rangle \approx 0$) yet highly correlated with the samples already in $\mathcal{S}$. Thus the optimization problem of Wang et al. (2024) is non-monotone, as adding an element may decrease the objective.

In the following section, we address these concerns by associating each sample $\mathbf{z} \in \mathcal{B}_t$ with a non-negative weight indicative of its importance. Introducing these importance weights alleviates the above issues and ensures that the resulting problem is a monotone, weakly submodular maximization problem.

## 4. Proposed Approach

We propose to select a subset $\mathcal{S} \subseteq \mathcal{B}_t$ that maximizes the following objective:

$$\max_{\mathcal{S} \subseteq \mathcal{B}_t, \, |\mathcal{S}| \leq \kappa} f(\mathcal{S}) \quad (4)$$

where $f(\mathcal{S})$ defines the utility of a set $\mathcal{S}$ as

$$f(\mathcal{S}) = \max_{\mathbf{w} \in \mathbb{R}_+^{|\mathcal{B}_t|}, \, \mathsf{supp}(\mathbf{w}) \subseteq \mathcal{S}} \mathcal{U}_t(\mathbf{w}, \mathcal{B}_t; \mathbf{z}_{\mathsf{val}}), \quad (5)$$

$$\mathcal{U}_t(\mathbf{w}, \mathcal{B}_t; \mathbf{z}_{\mathsf{val}}) = \langle \mathbf{w}, \boldsymbol{\mu}_{\boldsymbol{\theta}_t} \rangle - \frac{\eta_t}{2} \langle \mathbf{w}, \mathbf{K}_{\boldsymbol{\theta}_t} \mathbf{w} \rangle \quad (6)$$

where $\boldsymbol{\mu}_{\boldsymbol{\theta}_t} = \mathbf{G}_{\boldsymbol{\theta}_t} \boldsymbol{g}_{\boldsymbol{\theta}_t}(\mathbf{z}_{\mathsf{val}}) + \frac{\eta_t}{2} \boldsymbol{\nu}_{\boldsymbol{\theta}_t}$, $\mathbf{K}_{\boldsymbol{\theta}_t} = \mathbf{G}_{\boldsymbol{\theta}_t} \mathbf{G}_{\boldsymbol{\theta}_t}^\top$, $\boldsymbol{\nu}_{\boldsymbol{\theta}_t} = \mathrm{diag}(\mathbf{K}_{\boldsymbol{\theta}_t})$, $\mathbf{G}_{\boldsymbol{\theta}_t}$ denotes the batchwise gradient matrix over all the samples, i.e., $\mathbf{G}_{\boldsymbol{\theta}_t} = [\boldsymbol{g}_{\boldsymbol{\theta}_t}(\mathbf{z}_1), \ldots, \boldsymbol{g}_{\boldsymbol{\theta}_t}(\mathbf{z}_{|\mathcal{B}_t|})]^\top$, and $\mathrm{diag}(\cdot)$ denotes the vector of diagonal entries of a square matrix. See Appendix E.3 for our memory-efficient construction of per-example gradient features. We note that, given $\mathcal{S}$, computing $f(\mathcal{S})$ in (5) is a convex problem (concave maximization over a convex set) and can be solved optimally using gradient-based solvers. Moreover, as the per-sample LLM gradients are very high-dimensional ($d \gg |\mathcal{B}_t|$), the Gram matrix $\mathbf{K}_{\boldsymbol{\theta}_t} = \mathbf{G}_{\boldsymbol{\theta}_t} \mathbf{G}_{\boldsymbol{\theta}_t}^\top$ is positive definite ($\mathbf{K}_{\boldsymbol{\theta}_t} \succ 0$). Such properties make the proposed formulation amenable to efficient optimization techniques such as accelerated gradient descent and orthogonal matching pursuit (discussed in Section 6). We further note that $f$ is monotone non-decreasing: enlarging the support set $\mathcal{S}$ only enlarges the feasible region of the inner maximization, and $\mathcal{U}_t(\mathbf{0}, \mathcal{B}_t; \mathbf{z}_{\mathsf{val}}) = 0$.

The following remark shows that the GREATS optimization objective is a special case of our formulation.

*Remark* 4.1. Consider the setting in which the weights $\mathbf{w}$ are restricted to be binary, i.e., $\mathbf{w} \in \{0, 1\}^{|\mathcal{B}_t|}$, in (5). Then the marginal gain $\boldsymbol{\Delta} f(\mathbf{z} \mid \mathcal{S})$ coincides with the objective maximized by GREATS (Wang et al., 2024) in (3) .

**Domain-wise cardinality constraints and expected utility**. As discussed in Section 3, the data distribution is heterogeneous in real-world applications. To handle this, we (i) aggregate the utility over the validation set and (ii) enforce per-domain selection budgets. For (i), define the expected utility gain

$$\mathcal{U}_t(\mathbf{w}) := \mathbb{E}_{\mathbf{z}_{\mathsf{val}} \sim \mathcal{D}_{\mathsf{val}}} \mathcal{U}_t(\mathbf{w}, \mathcal{B}_t; \mathbf{z}_{\mathsf{val}}) \\ = \langle \mathbf{w}, \bar{\boldsymbol{\mu}}_{\boldsymbol{\theta}_t} \rangle - \frac{\eta_t}{2} \langle \mathbf{w}, \mathbf{K}_{\boldsymbol{\theta}_t} \mathbf{w} \rangle,$$

where $\mathbb{E}_{\mathbf{z}_{\mathsf{val}} \sim \mathcal{D}_{\mathsf{val}}}[\cdot]$ denotes the empirical average over the validation set $\mathcal{D}_{\mathsf{val}}$. Since the batchwise gradient matrix $\mathbf{G}_{\boldsymbol{\theta}_t}$ does not depend on $\mathbf{z}_{\mathsf{val}}$, the quadratic term is unchanged under the expectation, and only the linear term is averaged: $\bar{\boldsymbol{\mu}}_{\boldsymbol{\theta}_t} = \mathbf{G}_{\boldsymbol{\theta}_t} \bar{\boldsymbol{g}}_{\boldsymbol{\theta}_t} + \frac{\eta_t}{2} \boldsymbol{\nu}_{\boldsymbol{\theta}_t}$ with $\bar{\boldsymbol{g}}_{\boldsymbol{\theta}_t} := \mathbb{E}_{\mathbf{z}_{\mathsf{val}} \sim \mathcal{D}_{\mathsf{val}}}[\boldsymbol{g}_{\boldsymbol{\theta}_t}(\mathbf{z}_{\mathsf{val}})]$. Our subsequent analysis uses this aggregated utility $\mathcal{U}_t(\cdot)$ throughout.

For (ii), we enforce the per-domain budgets $\{\kappa_c^t\}_{c=1}^C$ through a *partition matroid* on the index set of $\mathcal{B}_t$, rather than through explicit sparsity constraints on $\mathbf{w}$. Construct $\mathcal{M}_t = (\mathcal{B}_t, \mathcal{I}_t)$ with the canonical domain partition

**Algorithm 1** PARTITIONSEL

1: **Input:** $T \in \mathbb{N}$ (total training steps)
2: **Output:** Final model parameters $\boldsymbol{\theta}_{T+1}$
3: Initialize model parameters $\boldsymbol{\theta}_0$
4: **for** $t = 1$ to $T$ **do**
5:     Receive mini-batch $\mathcal{B}_t = \biguplus_{c=1}^{C} \mathcal{B}_t^c$
6:     Set domain budgets $\{\kappa_c^t\}_{c=1}^{C}$; build partition matroid $\mathcal{M}_t$
7:     Form aggregated utility $\mathcal{U}_t(\cdot)$ over $\mathcal{B}_t$ (6)
8:     $\hat{\mathcal{S}}_t \leftarrow \text{OMP}(\mathcal{U}_t(\cdot), \mathcal{B}_t, \mathcal{M}_t)$     **(Alg. 2)**
9:     Update model using $\hat{\mathcal{S}}_t$
10: **end for**
11: **return** $\boldsymbol{\theta}_{T+1}$

**Algorithm 2** OMP: Weakly Submodular Maximization under a Matroid Constraint

1: **Input:** Objective $\mathcal{U}_t(\cdot)$, ground set $\mathcal{B}_t$, partition matroid $\mathcal{M}_t$ with capacities $\{\kappa_c^t\}_{c=1}^{C}$, budget $\kappa = \sum_{c=1}^{C} \kappa_c^t$
2: **Output:** Selected support $\Re_\kappa$
3: Initialize $\Re_0 \leftarrow \emptyset$, $\mathbf{w}_{\Re_0} \leftarrow \mathbf{0}$
4: $\boldsymbol{g} \leftarrow \nabla_{\mathbf{w}_{\Re_0}} \mathcal{U}_t(\mathbf{w}_{\Re_0}, \mathcal{B}_t)$
5: **for** $i = 1$ to $\kappa$ **do**
6:     Let $M_i$ be a maximal independent set of $\mathcal{M}/\Re_{i-1}$ maximizing $\sum_{\boldsymbol{x} \in M_i} \max(0, [\boldsymbol{g}]_{\boldsymbol{x}})$
7:     Sample $\boldsymbol{x}$ uniformly at random from $M_i$
8:     $\Re_i \leftarrow \Re_{i-1} \cup \{\boldsymbol{x}\}$
9:     $\mathbf{w}_{\Re_i} \leftarrow \arg\max_{\mathbf{w} \geq \mathbf{0}: \text{supp}(\mathbf{w}) \subseteq \Re_i} \mathcal{U}_t(\mathbf{w})$   **(APGA, Alg. 3)**
10:     $\boldsymbol{g} \leftarrow \nabla_{\mathbf{w}_{\Re_i}} \mathcal{U}_t(\mathbf{w}_{\Re_i})$
11: **end for**
12: **return** $\Re_\kappa$

$\mathcal{B}_t = \biguplus_{c=1}^{C} \mathcal{B}_t^c$ and per-block capacities $\{\kappa_c^t\}_{c=1}^{C}$, so that

$$\mathcal{S} \in \mathcal{I}(\mathcal{M}_t) \iff |\mathcal{S} \cap \mathcal{B}_t^c| \leq \kappa_c^t \; \forall c.$$

For any $\mathcal{A} \subseteq \mathcal{B}_t$, let $\bar{\mathcal{A}} \subseteq [|\mathcal{B}_t|]$ denote its index set; the map $\mathcal{A} \leftrightarrow \bar{\mathcal{A}}$ is a bijection. We define the index-set utility

$$f_{\mathcal{M}}(\bar{\mathcal{A}}) := \max_{\mathbf{w} \in \mathbb{R}_+^{|\mathcal{B}_t|}, \, \text{supp}(\mathbf{w}) \subseteq \bar{\mathcal{A}}} \mathcal{U}_t(\mathbf{w}), \qquad (7)$$

where the subscript $\mathcal{M}$ merely signals the matroid context and pose the domain-aware selection problem as

$$\max_{\mathcal{S} \in \mathcal{I}(\mathcal{M}_t)} f_{\mathcal{M}}(\bar{\mathcal{A}}). \qquad (8)$$

Crucially, no per-domain sparsity constraint is needed inside $f_{\mathcal{M}}$: matroid independence $\mathcal{S} \in \mathcal{I}(\mathcal{M}_t)$ together with $\text{supp}(\mathbf{w}) \subseteq \bar{\mathcal{A}}$ already forces $\|\mathbf{w}_{\mathcal{B}_t^c}\|_0 = |\text{supp}(\mathbf{w}) \cap \mathcal{B}_t^c| \leq |\mathcal{S} \cap \mathcal{B}_t^c| \leq \kappa_c^t$ for all $c$, so the budgets are enforced entirely by the matroid. Consequently, the inner program $f_{\mathcal{M}}$ is convex (concave maximization over the support-restricted non-negative orthant), and the only combinatorial structure resides in the outer matroid selection.

Let $\zeta(\bar{\mathcal{A}}) := \arg\max_{\mathbf{w} \geq \mathbf{0}, \, \text{supp}(\mathbf{w}) \subseteq \bar{\mathcal{A}}} \mathcal{U}_t(\mathbf{w})$ be the inner maximizer, which is unique since $\mathbf{K}_{\boldsymbol{\theta}_t} \succ 0$ makes $\mathcal{U}_t$ strictly concave, so $f_{\mathcal{M}}(\bar{\mathcal{A}}) = \mathcal{U}_t(\zeta(\bar{\mathcal{A}}))$. Thus maximizing $f_{\mathcal{M}}(\cdot)$ over independent index sets is equivalent to maximizing $\mathcal{U}_t(\cdot)$ over the corresponding partition-sparse non-negative cone.

## 5. Theoretical Results

We now state our key theoretical results; the proofs are provided in the supplementary material (Appendix C). The proof strategy relies on establishing the restricted strong concavity (RSC) and restricted smoothness (RSM) of the proposed objective. The relationship between RSC/RSM and weak submodularity has been studied in (Elenberg et al., 2018; Gurumoorthy et al., 2019), and our proofs build on those results.

Our first result shows that the proposed weighted mini-batch selection objective (7) is monotone. Hence, unlike (Wang et al., 2024), the marginal gain of any element is non-negative.

**Lemma 5.1.** *(Monotonicity) Let $f_{\mathcal{M}}(\cdot)$ be defined as in (7). Then for any index sets $\tilde{\mathcal{A}} \subseteq \tilde{\mathcal{B}}$ we have $f_{\mathcal{M}}(\tilde{\mathcal{A}}) \leq f_{\mathcal{M}}(\tilde{\mathcal{B}})$.*

**Lemma 5.2.** *(Lipschitz continuity of the gradient) For the utility function $\mathcal{U}_t(\mathbf{w}) = \langle \mathbf{w}, \boldsymbol{\mu}_{\boldsymbol{\theta}_t} \rangle - \frac{\eta_t}{2} \mathbf{w}^\top \mathbf{K}_{\boldsymbol{\theta}_t} \mathbf{w}$ defined in (6) with $\mathbf{K}_{\boldsymbol{\theta}_t} \succ 0$, the gradient $\nabla \mathcal{U}_t$ is Lipschitz continuous (in the Euclidean norm) with Lipschitz constant $\mathcal{L} = \eta_t \|\mathbf{K}_{\boldsymbol{\theta}_t}\|_{\text{op}} = \eta_t \|\mathbf{G}_{\boldsymbol{\theta}_t} \mathbf{G}_{\boldsymbol{\theta}_t}^\top\|_{\text{op}}$.*

**Lemma 5.3.** *(Finite RSC/RSM) Let $\mathbf{K}_{\boldsymbol{\theta}_t} \in \mathbb{R}^{|\mathcal{B}_t| \times |\mathcal{B}_t|}$ be symmetric positive definite and consider the concave quadratic $\mathcal{U}_t(\mathbf{w}) := \langle \mathbf{w}, \boldsymbol{\mu}_{\boldsymbol{\theta}_t} \rangle - \frac{\eta_t}{2} \mathbf{w}^\top \mathbf{K}_{\boldsymbol{\theta}_t} \mathbf{w}$. Then, for any sparsity level $s$, $\mathcal{U}_t(\cdot)$ is restricted strongly concave and restricted smooth on the domain $\Omega_s$ of $s$-sparse non-negative vectors, with parameters $c_s = \eta_t \lambda_{\min}^{(s)}(\mathbf{K}_{\boldsymbol{\theta}_t})$ and $C_s = \eta_t \lambda_{\max}^{(s)}(\mathbf{K}_{\boldsymbol{\theta}_t})$, where $\lambda_{\min}^{(s)}(\mathbf{K}) := \min_{\|\mathbf{v}\|_0 \leq s, \|\mathbf{v}\|_2 = 1} \mathbf{v}^\top \mathbf{K} \mathbf{v}$ and $\lambda_{\max}^{(s)}(\mathbf{K}) := \max_{\|\mathbf{v}\|_0 \leq s, \|\mathbf{v}\|_2 = 1} \mathbf{v}^\top \mathbf{K} \mathbf{v}$ are the extremal $s$-sparse (restricted) eigenvalues of $\mathbf{K}_{\boldsymbol{\theta}_t}$. In particular, since $\mathbf{K}_{\boldsymbol{\theta}_t} \succ 0$, we have $c_s > 0$ for every $s$.*

Our next result shows that the proposed weighted subset-selection function is weakly submodular; positivity of the submodularity ratio is a direct consequence of $\mathbf{K}_{\boldsymbol{\theta}_t} \succ 0$ (Lemma 5.3), which forces the restricted minimum eigenvalue, and hence the RSC constant, to be strictly positive.

**Lemma 5.4.** *The proposed function $f_{\mathcal{M}}(\cdot)$ in (7) is $\gamma$-weakly submodular with submodularity ratio $\gamma > 0$.*

Lemmas **5.1** and **5.4** together imply that PARTITIONSEL (Algorithm 1) enjoys the following approximation guarantee.

**Theorem 5.5.** *Let $\hat{\mathcal{S}}_t \in \mathcal{I}(\mathcal{M}_t)$ be the subset returned by* PARTITIONSEL *and let $\mathcal{S}^* \in \arg\max_{\mathcal{S} \in \mathcal{I}(\mathcal{M}_t)} f_{\mathcal{M}}(\bar{\mathcal{S}})$ be an optimal independent set under the partition matroid $\mathcal{M}_t$. Then*

$$\mathbb{E}\Big[f_{\mathcal{M}}(\hat{\mathcal{S}}_t)\Big] \geq (1 + \widetilde{C}_1/c_{2\kappa})^{-2} f_{\mathcal{M}}(\mathcal{S}^*),$$

*where the expectation is taken over the internal randomness of* PARTITIONSEL.

The proof follows from (Manupriya et al., 2024a, Lemma 3.3), where $c_{2\kappa}$ is the RSC constant of $\mathcal{U}_t(\cdot)$ on the $2\kappa$-sparse domain $\Omega_{2\kappa}$, and $\widetilde{C}_1$ is the RSM constant on $\widetilde{\Omega} = \{(\mathbf{w}, \mathbf{w}') : \|\mathbf{w} - \mathbf{w}'\|_0 \leq 1\}$ (see Section 2). Note that the approximation factor $(1 + \widetilde{C}_1/c_{2\kappa})^{-2}$ is invariant to $\eta_t$, since both $\widetilde{C}_1$ and $c_{2\kappa}$ scale linearly with $\eta_t$.

# 6. Algorithm

Algorithm 1 summarizes the training procedure for PARTITIONSEL. At each training step $t \in [T]$, we receive a mini-batch $\mathcal{B}_t$ partitioned into $C$ domains and specify per-domain cardinality (matroid) constraints $\{\kappa_c^t\}_{c=1}^C$. We then form the step-specific utility $\mathcal{U}_t$ over subsets of $\mathcal{B}_t$ and invoke the orthogonal matching pursuit (OMP) subroutine (Algorithm 2) to select a feasible support $\hat{\mathcal{S}}_t$ under these constraints. Finally, we update the model parameters using the selected subset $\hat{\mathcal{S}}_t$ and continue until step $T$.

**OMP subroutine.** Algorithm 2 approximately maximizes the weakly submodular utility $\mathcal{U}_t$ subject to the partition matroid $\mathcal{M}_t$ (encoding the domain-wise budgets) with total sparsity $\kappa = \sum_{c=1}^C \kappa_c^t$. Starting from an empty support $\Re_0$, OMP iteratively adds one element per round. At iteration $i$, it computes the gradient $\mathbf{g} = \nabla_{\mathbf{w}} \mathcal{U}_t(\mathbf{w}_{\Re_{i-1}})$, constructs a maximal independent set $M_i$ in the contracted matroid $\mathcal{M}_t/\Re_{i-1}$ that maximizes the aggregate positive gradient mass $\sum_{\boldsymbol{x} \in M_i} \max(0, [\mathbf{g}]_{\boldsymbol{x}})$, samples an element uniformly from $M_i$, and augments the support. Given the updated support $\Re_i$, it refits the weights by solving $\mathbf{w}_{\Re_i} \in \arg\max_{\mathbf{w} \geq \mathbf{0}: \mathrm{supp}(\mathbf{w}) \subseteq \Re_i} \mathcal{U}_t(\mathbf{w})$ (via APGA) and refreshes the gradient for the next round. After $\kappa$ rounds, the selected support $\Re_\kappa$ is returned.

In step 6 of Algorithm 2, $\mathcal{M}_t/\Re = (\mathcal{B}_t \setminus \Re, \mathcal{I}_{\mathcal{M}_t/\Re})$ denotes the contraction of $\mathcal{M}_t$ by $\Re$: a matroid on $\mathcal{B}_t \setminus \Re$ whose independent sets are $\mathcal{I}_{\mathcal{M}_t/\Re} := \{\mathcal{S} \subseteq \mathcal{B}_t \setminus \Re : \mathcal{S} \cup \Re \in \mathcal{I}_t\}$. We present the pseudocode for the accelerated projected gradient ascent (APGA) subroutine in Appendix F (Algo-

rithm 3).

**Complexity of Algorithm 2**: The per-iteration cost of Algorithm 2 is $\mathcal{O}(N + \tau M)$, where $N$ is the cost of computing the candidate gradients (line 10), $M$ is the gradient cost per APGA iteration (line 9), and $\tau$ is the number of APGA iterations. Since OMP runs $\kappa$ rounds, the total cost is $\mathcal{O}(\kappa(N + \tau M))$.

**Uniform Constraints.** We allocate the per-domain selection budget at iteration $t$ proportionally to domain size, $\kappa_c^t = \lfloor \kappa |\mathcal{B}_t^c|/|\mathcal{B}_t| \rfloor$, so that each domain $c$ receives a budget in proportion to its representation in the mini-batch (rounded so that $\sum_{c=1}^C \kappa_c^t = \kappa$).

## 6.1. Independent Domain wise selection

We also consider an independent domain-selection baseline based on (Wang et al., 2024), in which each domain is allotted a budget $\kappa_c^t$ and selected separately. The resulting objective is

$$f^{\mathsf{ID}}(\mathcal{S}) = \sum_{c=1}^C \max_{\mathbf{w}_c \in \{0,1\}^{|\mathcal{B}_t^c|},\ \|\mathbf{w}_c\|_0 \leq \kappa_c^t} \mathcal{U}_t(\mathbf{w}_c, \mathcal{B}_t^c; \mathbf{z}_{\mathsf{val}}).$$

where the overall selection is $\mathcal{S} = \bigcup_{c=1}^C \mathrm{supp}(\mathbf{w}_c)$. We term this baseline ID. Each domain-specific selection vector $\mathbf{w}_c$ is binary-valued, inducing a hard subset selection within each domain. We additionally consider a relaxed variant (IWD) that performs domain-wise independent selection with continuous prototype weights, enabling soft weighting rather than combinatorial subset selection ($\mathbf{w}_c$ continuous):

$$f^{\mathsf{IWD}}(\mathcal{S}) = \sum_{c=1}^C \max_{\mathbf{w}_c \in \mathbb{R}_+^{|\mathcal{B}_t^c|},\ \|\mathbf{w}_c\|_0 \leq \kappa_c^t} \mathcal{U}_t(\mathbf{w}_c, \mathcal{B}_t^c; \mathbf{z}_{\mathsf{val}}) \quad (9)$$

again with $\mathcal{S} = \bigcup_{c=1}^C \mathrm{supp}(\mathbf{w}_c)$.

## 6.2. Is PARTITIONSEL a Scalable Proxy for Continuous Data Mixture Methods?

**DoReMi Objective**. Continuous data mixture methods such as DoReMi (Xie et al., 2023a) compute optimal domain weights by explicitly optimizing a minimax objective over domains. Formally, DoReMi solves

$$\min_{\boldsymbol{\theta}} \max_{\boldsymbol{\alpha} \in \Delta^C} \mathcal{L}(\boldsymbol{\theta}, \boldsymbol{\alpha}) := \sum_{c=1}^C \boldsymbol{\alpha}_c \frac{1}{|\mathcal{V}_c|} \sum_{\boldsymbol{x} \in \mathcal{V}_c} (\ell_{\boldsymbol{\theta}}(\boldsymbol{x}) - \ell_{\mathrm{ref}}(\boldsymbol{x})).$$

The optimal $\boldsymbol{\alpha}$ defines a continuous mixture distribution over training data, $\Pi_{\boldsymbol{\alpha}} = \sum_{c=1}^C \boldsymbol{\alpha}_c \frac{1}{|\mathcal{V}_c|} \sum_{\boldsymbol{x} \in \mathcal{V}_c} \delta_{\boldsymbol{x}}$, which upweights domains with higher excess loss $\ell_{\boldsymbol{\theta}}(\boldsymbol{x}) - \ell_{\mathrm{ref}}(\boldsymbol{x})$. In practice this objective is optimized via Group Distributionally Robust Optimization (Group DRO) (Sagawa et al., 2019): domains exhibiting higher excess loss are

upweighted, increasing their influence on subsequent gradient updates. The resulting effective data mixture is the time-averaged iterate $\bar{\boldsymbol{\alpha}} = \sum_{t=1}^{T} \boldsymbol{\alpha}_t / T$, which is then used to train the final main model (parameterized by $\boldsymbol{\theta}_m$):

$$\min_{\boldsymbol{\theta}_m} \Big[ \sum_{c=1}^{C} \bar{\boldsymbol{\alpha}}_c \Big( \frac{1}{|\mathcal{V}_c|} \sum_{\boldsymbol{x} \in \mathcal{V}_c} \ell_{\boldsymbol{\theta}_m}(\boldsymbol{x}) \Big) \Big].$$

Crucially, DoReMi represents domain importance through a *continuous* reweighting of the data distribution. This raises the question of whether similar information can be captured using only *discrete* sample selection. To this end, we view PARTITIONSEL as inducing an implicit domain mixture through its per-step selected subset $\hat{\mathcal{S}}_t$.

PARTITIONSEL. For a direct comparison, we define the induced domain weight of PARTITIONSEL at iteration $t$ as $\boldsymbol{\alpha}_{c,t}^{\text{DOMW}} = \left| \hat{\mathcal{S}}_t \cap \mathcal{V}_c \right| / \left| \hat{\mathcal{S}}_t \right|$, which measures the empirical mass assigned to domain $c$ by the selected samples. We denote by $\boldsymbol{\alpha}^{\text{DORM}} := \bar{\boldsymbol{\alpha}}$ the domain mixture obtained by DoReMi. This construction lets us interpret PARTITIONSEL as a discrete, sample-level approximation to the continuous domain reweighting. Figure 1 and Table 3 indicate the effectiveness of our method.

# 7. Experimental Results

**Baselines** We compare our method PARTITIONSEL against the following baselines (i) **GREATS** (Wang et al., 2024), (ii) **ID**: An independent domain selection baseline. While GREATS objective have a single overall cardinality constraint across all the domains, ID selects mini-batches from each domain independently using GREATS (iii) **IWD** as defined in Section 6.1 (iv) **COLM** (Nguyen et al., 2025) (v) **GradNorm** (Katharopoulos & Fleuret, 2018) **Random**: Selects mini-batches randomly.

## 7.1. Standard Finetuning

**Models and Training Details.** We evaluate on QWEN2.5 (3B and 7B) (Team, 2024), LLAMA3 (3B and 8B) (Dubey et al., 2024). For finetuning, we employ LoRA with rank 16, $\alpha = 96$, and dropout 0.05. Following (Nguyen et al., 2025), LoRA is applied to all attention matrices (QKV$_{\text{proj}}$) and two fully connected layers, with a learning rate of 2e-5.

**Finetuning Datasets.** We train on the METAMATHQA dataset (Yu et al., 2024a), which consists of 260K instruction tuning samples curated from 14 highly imbalanced data sources. Please see Appendix D.2 for detailed description.

**Evaluation datasets.** Following (Yue et al., 2024), we evaluate the finetuned models on both in-domain and out-of-domain benchmarks. The in-domain set comprises GSM8K (Cobbe et al., 2021), MATH (Hendrycks et al., 2021b), and NumGLUE (Mishra et al., 2022), whereas the out-of-domain set includes SVAMP (Patel et al., 2021), Mathematics (Davies et al., 2021), SimulEq (Koncel-Kedziorski et al., 2016), MMLU (Hendrycks et al., 2021a), and AQuA (Ling et al., 2017).

**Results on finetuning for Math-reasoning:** We train all models for 2048 steps with batch size of 128 and prototype-ratio as 50% (thus effective batch size $\kappa = 64$). Table 1 shows the results of source-wise fine-tuning on META-MATHQA. The table indicates that PARTITIONSEL is significantly better than other baselines. Additional results for QWEN2.5-7B, LLAMA-3.2-3B, and LLAMA-3.1-8B are reported in the supplementary material (Figure 9 and Tables 9 to 11).

## 7.2. Molecule generation Experiments

**Finetuning Datasets.** To determine the effectiveness of PARTITIONSEL on out-of-domain datasets, we inspect baselines on the task of Molecule generation. Roughly, the model must generate valid and parseable molecule in the SMILES (Weininger, 1988) format, given an instruction like:

```
Design a protein which possesses
methylglyoxal synthase activity.
```

Please see the Appendix D.1 for a detailed example. We utilise MOL-INSTRUCTIONS (Fang et al., 2023) and evaluate on four domains:

1. **Description-guided design:** generate a valid molecule (SMILES) from a natural-language functional description.

2. **Forward reaction prediction:** predict the reaction product given reactants and reagents.

3. **Reagent prediction:** predict auxiliary reagents given reactants and product.

4. **Single-step retrosynthesis:** infer feasible reactants given a target product.

**Models and Training Details.** We finetune QWEN2.5-3B using LoRA (Hu et al., 2022) of rank 16 and alpha 96, using a learning rate of 2e-4 for 1024 steps using *Adam* (Kingma & Ba, 2015). Following the experimental setup in (Wang et al., 2024), we use a validation batch of size 2 to guide the minibatch selection. We split the dataset into validation and test datasets, with each domain containing 1000 examples. We set the $\hat{\mathcal{S}}_t$, the full-batch size to be 64 for all experiments.

**Results and discussion.** Table 2 shows the BLEU score and Edit-distance of baselines on (Fang et al., 2023). Table 2 indicates that PARTITIONSEL is consistently better than the baselines on both metrics and at varying selection fraction $\kappa$.

*Table 1.* Downstream results on math-reasoning benchmarks after fine-tuning QWEN2.5-3B on METAMATHQA (subset fraction=50%).

| Method | NumGLUE | MMLU-Math | GSM8K | SVAMP | SimulEq | DeepMind | AQuA | SAT | Avg |
|---|---|---|---|---|---|---|---|---|---|
| Random | 0.5704 | 0.5332 | 0.7771 | 0.832 | 0.5027 | 0.3281 | 0.6086 | 0.6594 | 0.6014 |
| GradNorm (Katharopoulos & Fleuret, 2018) | **0.6084** | 0.5565 | 0.7612 | 0.8410 | 0.5156 | 0.3340 | 0.5945 | 0.6727 | 0.6105 |
| COLM (Nguyen et al., 2025) | 0.5988 | 0.5503 | 0.7627 | 0.8420 | 0.5156 | 0.3570 | 0.5827 | 0.6864 | 0.6119 |
| GREATS (Wang et al., 2024) | 0.5873 | **0.6109** | 0.7892 | 0.8480 | 0.5000 | **0.3700** | 0.6142 | 0.6727 | 0.6240 |
| ID | 0.5960 | 0.5965 | **0.7771** | **0.8540** | 0.5117 | 0.3700 | 0.5866 | 0.7182 | 0.6263 |
| IWD | 0.6115 | 0.5795 | 0.7699 | 0.8500 | 0.5218 | 0.3675 | 0.5980 | 0.6818 | 0.6225 |
| PARTITIONSEL (*Ours*) | 0.5998 | 0.6099 | 0.7665 | 0.8410 | **0.5467** | 0.3500 | **0.6220** | **0.7545** | **0.6363** |

*Table 2.* Performance by training QWEN2.5-3B on MOL-INSTRUCTIONS (Fang et al., 2023) with $|\hat{S}_t| = 128$, evaluated using BLEU / Edit-distance. Reported at $\kappa = 16 / 32$, meaning subset-fractions of 12.5 / 25%. For BLEU **higher is better**. For Edit-distance, **lower is better**. PARTITIONSEL consistently beats baselines.

| Baseline | Desc. guided design | | Forward reaction pred. | | Reagent pred. | | Retrosynthesis | | Average | |
|---|---|---|---|---|---|---|---|---|---|---|
| | 12.5% | 25% | 12.5% | 25% | 12.5% | 25% | 12.5% | 25% | 12.5% | 25% |
| Random | 0.37 / 47.69 | | 0.83 / 19.76 | | 0.49 / 31.78 | | 0.81 / 23.83 | | 0.62 / 30.77 | |
| GradNorm | 0.27 / 48.30 | 0.32 / **45.70** | 0.67 / 23.09 | 0.73 / 21.21 | 0.19 / 65.40 | 0.23 / 58.62 | 0.74 / 28.33 | 0.78 / 26.92 | 0.47 / 41.28 | 0.52 / 38.11 |
| COLM (Nguyen et al., 2025) | 0.34 / 49.30 | 0.28 / 46.90 | **0.84** / 19.37 | 0.83 / **19.57** | 0.48 / **31.10** | 0.46 / 32.27 | 0.81 / **23.39** | 0.81 / 23.80 | 0.62 / **30.79** | 0.59 / 30.64 |
| GREATS (Wang et al., 2024) | 0.34 / 50.93 | 0.35 / 46.96 | 0.83 / 19.73 | 0.82 / 20.56 | 0.48 / 31.84 | 0.50 / 31.37 | 0.81 / 24.01 | 0.81 / 24.48 | 0.61 / 31.63 | 0.62 / 30.84 |
| ID | 0.35 / **47.59** | 0.36 / 46.70 | 0.83 / 20.09 | 0.82 / 20.50 | **0.49** / 31.79 | 0.47 / 32.50 | 0.81 / 24.67 | 0.81 / 24.10 | 0.62 / 31.04 | 0.61 / 30.95 |
| IWD | 0.38 / 48.25 | **0.42** / 47.60 | 0.83 / 19.73 | 0.83 / 19.58 | 0.48 / 32.92 | 0.47 / 33.32 | 0.81 / 24.39 | 0.81 / 23.76 | 0.62 / 31.32 | 0.63 / 31.07 |
| PARTITIONSEL (*Ours*) | **0.39** / 49.15 | **0.42** / 47.27 | **0.84** / 19.44 | **0.84** / 19.60 | 0.48 / 32.03 | **0.54** / **28.29** | **0.82** / 23.44 | **0.82** / **23.70** | **0.63** / 31.02 | **0.66** / **29.71** |

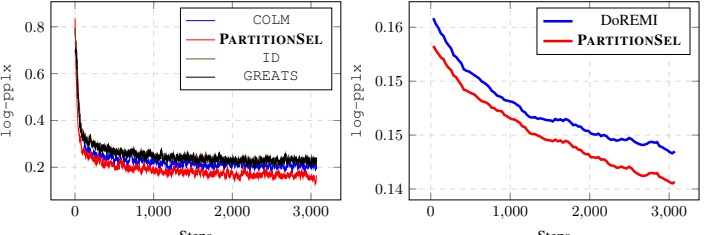

*Table 3.* Comparison of data mixture strategies

| Dataset | DoReMi | PARTITIONSEL | Δ |
|---|---|---|---|
| MMLU-Math | 53.08 | 60.57 | +7.49 |
| AQuA | 58.66 | 61.02 | +2.36 |
| SAT | 72.27 | 76.36 | +4.09 |
| **Average** | **61.34** | **65.99** | **+4.65** |

*Figure 1.* Validation `log-pplx` on METAMATHQA (Left), comparison with DoReMi in terms of validation `log-pplx` (Middle), and downstream benchmark performance (Right).

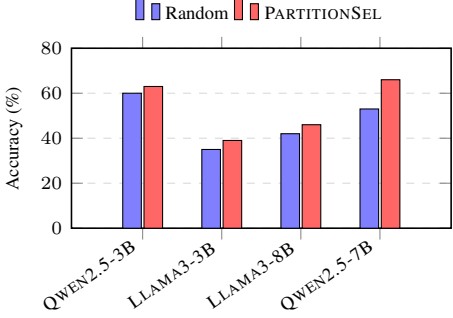

*Figure 2.* Comparing RANDOM selection with PARTITIONSEL for QWEN2.5 and LLAMA3 models for downstream accuracy.

### 7.3. Mixture methods

As discussed in Section 6.2, to compare with offline mixture methods like (Xie et al., 2023a), we utilise the baseline and proxy model as GPT2-300M (Radford et al., 2019), and finetune on METAMATHQA (Yu et al., 2024a), using

LoRA rank 16, alpha 32, dropout 0.05 for 1 epoch using *Adam* (Kingma & Ba, 2015). After training the DoReMi proxy, the learned domain mixture deviates sharply from the uniform initialization. Concretely, while the proxy is trained with uniform assigned weights ($\alpha_c = 0.125$ for all $c$ in 8 domains), the learned mixture concentrates most mass on MATH_Rephrased (0.608), with secondary weight on GSM_FOBAR (0.177) and MATH_FOBAR (0.076); all other domains receive comparatively small mass (Fig. 3). Fig. 1 indicates that PARTITIONSEL has consistently lower validation loss and better downstream performance as compared with DoReMi.

### 7.4. Reducing Gradient Conflicts across Domains

In multi-domain training, different domains can induce gradients that point in competing directions, slowing convergence and degrading transfer across tasks. This issue is amplified under subset selection, where the selected minibatch can become overly specialized to a single domain or

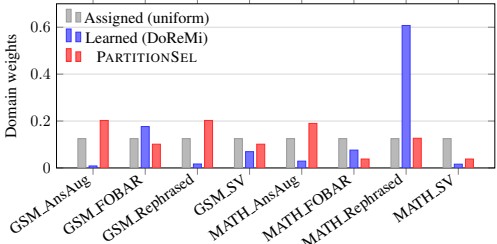

*Figure 3.* DoReMi: uniform assigned weights vs learned domain mixture after 1 epoch of baseline and proxy training.

objective. We compare two selection strategies: (i) `ID` that

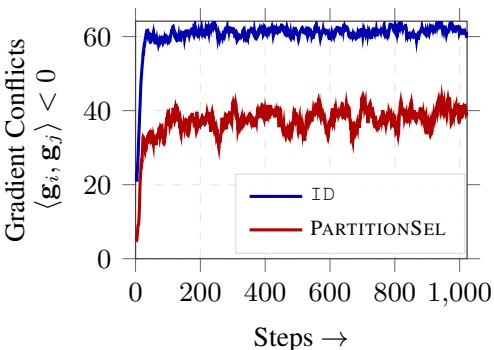

*Figure 4.* **Reducing gradient conflicts.** `ID` ignores cross-domain redundancy and can pick mutually conflicting gradients. PARTITIONSEL trades off samples across domains, improving alignment and reducing destructive interference.

selects samples separately within each domain and then concatenates them, and (ii) our *joint* selection that coordinates choices across domains under a global budget. Figure 4 illustrates the resulting reduction in conflicting gradient pairs and a more stable optimization trajectory.

## 8. Ablations

### 8.1. Effect of Randomised FFT

We measure the effect of performing Randomised FFT (Ailon & Chazelle, 2006) on the gradients to reduce inter-GPU memory traffic. Table 4 shows that FJLT enables gradient computation across more layers while providing theoretical guarantees on preserving pairwise inner products, the key quantity in our utility computation, making full high-dimensional gradients unnecessary.

### 8.2. Effect of number of layers for Gradients

To check the effect of finding gradients from more than 1 layer, we experiment with QWEN2.5-2B, LoRA rank 64, alpha 128. Figure 5 indicates that using more than 1 layers implies an improvement in validation loss, but comes with a relative performance penalty to just using the final layer.

*Table 4.* Feasibility of per-example gradient computation with and without FJLT (Qwen2.5-3B, LoRA rank 64).

| Method | Raw | FJLT |
|---|---|---|
| Grads-1 layer | ✓ | ✓ |
| Grads-2 layers | ✓ | ✓ |
| Grads-3 layers | OOM | ✓ |
| Grads-5 layers | OOM | ✓ |

### 8.3. Comparisons with different Models

To check the robustness of gains, we evaluate PARTITIONSEL across QWEN2.5 (3B and 7B) and LLAMA3 (3B and 8B). Figure 2 shows that PARTITIONSEL shows consistent gains for subset selection.

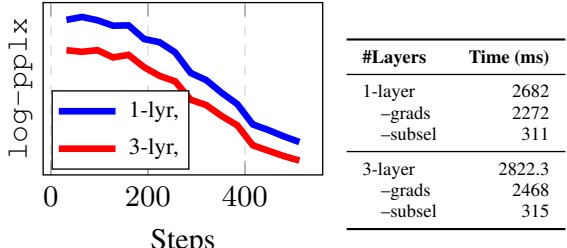

| #Layers | Time (ms) |
|---|---|
| 1-layer | 2682 |
| –grads | 2272 |
| –subsel | 311 |
| 3-layer | 2822.3 |
| –grads | 2468 |
| –subsel | 315 |

*Figure 5.* Latency breakdown across 1 layer vs 3-layer gradient computation.

## 9. Conclusion

We introduced PARTITIONSEL that explicitly optimizes domain representations under partition matroid constraints. By selecting samples jointly across domains using a validation-guided gradient matching objective, our method avoids the limitations of independent per-domain selection and expensive proxy-based weighting schemes. We showed that the resulting objective is weakly submodular, enabling an efficient OMP algorithm with theoretical guarantees. Empirically, our approach consistently improves performance, while reducing gradient conflicts during training. demonstrating it as an effective principled strategy for training across diverse domains.

## Acknowledgements

PC acknowledges the Microsoft Research India PhD Award and Prime Minister Research Fellowship to support this research work. GR thanks Bank of Baroda Chair Professorship. PJ acknowledges the support of SBI Foundation Hub for Data & Analytics grant and Anusandhan National Research Foundation for ARG-MATRICS grant.

## Impact Statement

This work develops a framework for prototype-selection aiming to select points jointly across domains in multi-domain settings. There are many positive societal consequences of our work, none which we feel must be specifically highlighted here.

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

# Supplementary Material: Minibatch selection for Language Models via Partition Matroid Constrained Gradient Matching

## A. Organization of Appendix

The Appendix is structured as follows. We dicuss some additional related works and background in Section B. Section C summarizes the main theoretical results. Section F presents the detailed description of the proposed algorithm. Sections E and E.5 outline the implementation aspects and the specifics of the experimental setup, respectively. Finally, Section H provides a link to the publicly available codebase.

## B. Background and additional related works

### B.1. Submodular maximization under matroid constraints

Our analysis builds on a long line of work on (weakly) submodular maximization. The classical greedy algorithm achieves a $(1 - 1/e)$ approximation for monotone submodular maximization under cardinality (Nemhauser et al., 1978) and a $1/2$ guarantee under matroid constraints, improved to $1 - 1/e$ by continuous-greedy / pipage rounding (Calinescu et al., 2011; Fisher et al., 1978). Das & Kempe (2018) introduce the submodularity ratio for non-submodular functions, and Elenberg et al. (2018) relate restricted strong concavity to weak submodularity, yielding approximation guarantees for orthogonal matching pursuit (Bian et al., 2017; Gurumoorthy et al., 2019). We leverage these tools to obtain provable guarantees for our partition-matroid-constrained, weakly submodular objective; to our knowledge, this is the first application of partition-matroid-constrained weakly submodular maximization to domain-aware LLM mini-batch selection.

### B.2. Approximate Submodularity

We adopt the notion of *approximate submodularity* characterized by the *submodularity ratio* as shown in (Das & Kempe, 2018). For a monotone set function $f$, the submodularity ratio with respect to a ground set $\mathcal{V}$, a reference set $\mathcal{S} \subseteq \mathcal{V}$, and a parameter $\kappa \geq 1$ is defined as

$$\alpha_{\mathcal{V},\kappa}(f) = \min_{\substack{\mathcal{S} \subseteq \mathcal{V}, \ \mathcal{A}: |\mathcal{A}| \leq \kappa, \\ \mathcal{A} \cap \mathcal{S} = \varnothing}} \frac{\sum_{u \in \mathcal{A}}(f(\mathcal{S} \cup \{u\}) - f(\mathcal{S}))}{f(\mathcal{S} \cup \mathcal{A}) - f(\mathcal{S})}, \qquad \text{with } 0/0 := 1.$$

The function $f(\cdot)$ is submodular if and only if $\alpha_{\mathcal{V},\kappa}(f) \geq 1$. More generally, if

$$\alpha \equiv \frac{\sum_{u \in \mathcal{A}}(f(\mathcal{S} \cup \{u\}) - f(\mathcal{S}))}{f(\mathcal{S} \cup \mathcal{A}) - f(\mathcal{S})} > 0,$$

but not necessarily $\alpha \geq 1$, then $f$ is said to be $\alpha$-*weakly submodular*.

### B.3. Conflict-aware multi-task and multi-domain optimization

Training on heterogeneous sources frequently produces conflicting per-task gradients, motivating methods that project, scale, or reweight updates to mitigate interference (Yu et al., 2020; Liu et al., 2021). Whereas these approaches modify the *optimizer* at the gradient level, PARTITIONSEL addresses gradient conflict at the *data-selection* level: its quadratic gradient-similarity term penalizes redundant or interfering samples within each mini-batch, which we observe empirically reduces the fraction of conflicting gradient pairs.

### B.4. High-quality instruction tuning and alignment data

Finally, several studies show that careful curation of instruction-tuning data can yield strong performance even with comparatively small datasets (Zhou et al., 2023; Yue et al., 2024), motivating principled mixture and filtering strategies for supervised fine-tuning (Renduchintala et al., 2024; Liu et al., 2024a) and for preference/alignment training (Deng et al., 2025; Tunstall et al., 2023).

# C. Theoretical Results

**Lemma 5.1.** *(Monotonicity) Let $f_{\mathcal{M}}(\cdot)$ be defined as in (7). Then for any index sets $\tilde{A} \subseteq \tilde{B}$ we have $f_{\mathcal{M}}(\tilde{A}) \leq f_{\mathcal{M}}(\tilde{B})$.*

*Proof.* Let $|\tilde{A}| = n_1$ and $|\tilde{B}| = n_2$ and since $\tilde{A} \subseteq \tilde{B}$ we have $n_1 < n_2$. We index the elements in $\tilde{B}$ such that the first $n_1$ elements are contained in $\tilde{A}$.

$$f_{\mathcal{M}}(\tilde{B}) = \max_{\mathbf{w} \in \mathbb{R}_+^{n_2}, supp(\mathbf{w}) \subseteq \tilde{B}} \mathcal{U}_t(\mathbf{w}) \geq \max_{\mathbf{w} \in \mathbb{R}_+^{n_1}, supp(\mathbf{w}) \subseteq \tilde{A}} \mathcal{U}_t(\mathbf{w}) = f_{\mathcal{M}}(\tilde{A}).$$

$\square$

**Lemma 5.2.** *(Lipschitz continuity of the gradient) For the utility function $\mathcal{U}_t(\mathbf{w}) = \langle \mathbf{w}, \boldsymbol{\mu}_{\boldsymbol{\theta}_t} \rangle - \frac{\eta_t}{2} \mathbf{w}^{\mathsf{T}} \mathbf{K}_{\boldsymbol{\theta}_t} \mathbf{w}$ defined in (6) with $\mathbf{K}_{\boldsymbol{\theta}_t} \succ \mathbf{0}$, the gradient $\nabla \mathcal{U}_t$ is Lipschitz continuous (in the Euclidean norm) with Lipschitz constant $\mathcal{L} = \eta_t \|\mathbf{K}_{\boldsymbol{\theta}_t}\|_{\mathrm{op}} = \eta_t \|\mathbf{G}_{\boldsymbol{\theta}_t} \mathbf{G}_{\boldsymbol{\theta}_t}^{\mathsf{T}}\|_{\mathrm{op}}$.*

*Proof.* Since $\mathcal{U}_t(\mathbf{w}) = \mathbf{w}^{\mathsf{T}} \boldsymbol{\mu}_{\boldsymbol{\theta}_t} - \frac{1}{2} \mathbf{w}^{\mathsf{T}} \mathbf{K}_{\boldsymbol{\theta}_t} \mathbf{w}$, differentiating with respect to $\mathbf{w}$ gives $\nabla \mathcal{U}_t(\mathbf{w}) = \boldsymbol{\mu}_{\boldsymbol{\theta}_t} - \mathbf{K}_{\boldsymbol{\theta}_t} \mathbf{w}$.

Given a batch $\mathcal{B}_t$, for any $\mathbf{w}, \mathbf{w}' \in \mathbb{R}^{|\mathcal{B}_t|}$, by the definition of the operator norm,

$$\|\nabla \mathcal{U}_t(\mathbf{w}) - \nabla \mathcal{U}_t(\mathbf{w}')\|_2 = \|\mathbf{K}_{\boldsymbol{\theta}_t}(\mathbf{w} - \mathbf{w}')\|_2 \leq \|\mathbf{K}_{\boldsymbol{\theta}_t}\|_{\mathrm{op}} \|\mathbf{w} - \mathbf{w}'\|_2,$$

so $\nabla \mathcal{U}_t$ is Lipschitz continuous with constant $\mathcal{L} = \|\mathbf{K}_{\boldsymbol{\theta}_t}\|_{\mathrm{op}}$.

Since $\mathbf{K}_{\boldsymbol{\theta}_t} \succ \mathbf{0}$ is symmetric positive definite it admits a factorisation $\mathbf{K}_{\boldsymbol{\theta}_t} = \mathbf{G}_{\boldsymbol{\theta}_t} \mathbf{G}_{\boldsymbol{\theta}_t}^{\mathsf{T}}$, and because the operator norm satisfies $\|\mathbf{A}\|_{\mathrm{op}} = \sup_{\|\mathbf{v}\|_2=1} \|\mathbf{A}\mathbf{v}\|_2$ for any matrix $\mathbf{A}$, we have directly $\|\mathbf{K}_{\boldsymbol{\theta}_t}\|_{\mathrm{op}} = \|\mathbf{G}_{\boldsymbol{\theta}_t} \mathbf{G}_{\boldsymbol{\theta}_t}^{\mathsf{T}}\|_{\mathrm{op}}$.

The constant $\mathcal{L} = \|\mathbf{K}_{\boldsymbol{\theta}_t}\|_{\mathrm{op}}$ is sharp: taking $\mathbf{w} - \mathbf{w}'$ to be the leading eigenvector $\mathbf{v}_{\max}$ of $\mathbf{K}_{\boldsymbol{\theta}_t}$ (with largest eigenvalue $\lambda_{\max}$) yields $\|\nabla \mathcal{U}_t(\mathbf{w}) - \nabla \mathcal{U}_t(\mathbf{w}')\|_2 = \lambda_{\max} \|\mathbf{w} - \mathbf{w}'\|_2 = \|\mathbf{K}_{\boldsymbol{\theta}_t}\|_{\mathrm{op}} \|\mathbf{w} - \mathbf{w}'\|_2$, confirming that no smaller constant suffices. $\square$

**Lemma 5.3.** *(Finite RSC/RSM) Let $\mathbf{K}_{\boldsymbol{\theta}_t} \in \mathbb{R}^{|\mathcal{B}_t| \times |\mathcal{B}_t|}$ be symmetric positive definite and consider the concave quadratic $\mathcal{U}_t(\mathbf{w}) := \langle \mathbf{w}, \boldsymbol{\mu}_{\boldsymbol{\theta}_t} \rangle - \frac{\eta_t}{2} \mathbf{w}^{\mathsf{T}} \mathbf{K}_{\boldsymbol{\theta}_t} \mathbf{w}$. Then, for any sparsity level $s$, $\mathcal{U}_t(\cdot)$ is restricted strongly concave and restricted smooth on the domain $\Omega_s$ of $s$-sparse non-negative vectors, with parameters $c_s = \eta_t \lambda_{\min}^{(s)}(\mathbf{K}_{\boldsymbol{\theta}_t})$ and $C_s = \eta_t \lambda_{\max}^{(s)}(\mathbf{K}_{\boldsymbol{\theta}_t})$, where $\lambda_{\min}^{(s)}(\mathbf{K}) := \min_{\|\mathbf{v}\|_0 \leq s, \|\mathbf{v}\|_2=1} \mathbf{v}^{\mathsf{T}} \mathbf{K} \mathbf{v}$ and $\lambda_{\max}^{(s)}(\mathbf{K}) := \max_{\|\mathbf{v}\|_0 \leq s, \|\mathbf{v}\|_2=1} \mathbf{v}^{\mathsf{T}} \mathbf{K} \mathbf{v}$ are the extremal $s$-sparse (restricted) eigenvalues of $\mathbf{K}_{\boldsymbol{\theta}_t}$. In particular, since $\mathbf{K}_{\boldsymbol{\theta}_t} \succ \mathbf{0}$, we have $c_s > 0$ for every $s$.*

*Proof.* The function $\mathcal{U}_t(\mathbf{w}) = \mathbf{w}^{\mathsf{T}} \boldsymbol{\mu}_{\boldsymbol{\theta}_t} - \frac{1}{2}\mathbf{w}^{\mathsf{T}} \mathbf{K}_{\boldsymbol{\theta}_t} \mathbf{w}$ is concave. We calculate $\mathcal{U}_t(\mathbf{w}_1) - \mathcal{U}_t(\mathbf{w}_2) - \nabla \langle \mathcal{U}_t(\mathbf{w}_2), \mathbf{w}_1 - \mathbf{w}_2 \rangle = -0.5(\mathbf{w}_1 - \mathbf{w}_2)^T \mathbf{K}_{\boldsymbol{\theta}_t}(\mathbf{w}_1 - \mathbf{w}_2)$. Since $\mathbf{w}$ is $k_1 + k_2$ sparse, we consider $\mathbf{w}_1$ as $k_1^1 + k_2^1$ and $\mathbf{w}_2$ as $k_1^2 + k_2^2$ sparse vectors respectively. Now, $\Delta \mathbf{w} = \mathbf{w}_1 - \mathbf{w}_2$ has a maximum of $k \leq k_1^1 + k_2^1 + k_1^2 + k_2^2$ non-zero entries. For the constants $b$ and $B$ satisfying $-b\|\Delta \mathbf{w}\|^2 \geq -\Delta \mathbf{w}^T \mathbf{K}_{\boldsymbol{\theta}_t} \Delta \mathbf{w} \geq -B\|\Delta \mathbf{w}\|^2$, we have $b \geq k$-sparse smallest eigenvalue of $\mathbf{K}_{\boldsymbol{\theta}_t}$ and $B \leq k$-sparse largest eigenvalue of $\mathbf{K}_{\boldsymbol{\theta}_t}$. $\square$

**Lemma 5.4.** *The proposed function $f_{\mathcal{M}}(\cdot)$ in (7) is $\gamma$-weakly submodular with submodularity ratio $\gamma > 0$.*

*Proof.* Fix arbitrary *disjoint* index sets $\mathcal{S}, \mathcal{A} \subseteq [|\mathcal{B}_t|]$ and let $\bar{\kappa} = |\mathcal{S}| + |\mathcal{A}| = |\mathcal{S} \cup \mathcal{A}|$. For an index set $\mathcal{T}$, let $\boldsymbol{\zeta}^{(\mathcal{T})} := \arg\max_{\mathbf{w} \geq \mathbf{0}, \, supp(\mathbf{w}) \subseteq \mathcal{T}} \mathcal{U}_t(\mathbf{w}) \in \mathbb{R}_+^{|\mathcal{B}_t|}$ denote the inner maximizer (cf. (7)), so that $f_{\mathcal{M}}(\mathcal{T}) = \mathcal{U}_t(\boldsymbol{\zeta}^{(\mathcal{T})})$. We bound the submodularity ratio

$$\gamma_{\mathcal{S}, \mathcal{A}} = \frac{\sum_{j \in \mathcal{A}}[f_{\mathcal{M}}(\mathcal{S} \cup \{j\}) - f_{\mathcal{M}}(\mathcal{S})]}{f_{\mathcal{M}}(\mathcal{S} \cup \mathcal{A}) - f_{\mathcal{M}}(\mathcal{S})}$$

from below by lower-bounding the numerator and upper-bounding the denominator.

**Upper bound on the denominator.** By restricted strong concavity (Section 2) applied to $\boldsymbol{x} = \boldsymbol{\zeta}^{(\mathcal{S})}$ and $\mathbf{y} = \boldsymbol{\zeta}^{(\mathcal{S} \cup \mathcal{A})}$ (whose difference is supported on $\mathcal{S} \cup \mathcal{A}$, hence $\bar{\kappa}$-sparse),

$$\frac{c_{\bar{\kappa}}}{2}\|\boldsymbol{\zeta}^{(\mathcal{S} \cup \mathcal{A})} - \boldsymbol{\zeta}^{(\mathcal{S})}\|^2 \leq \mathcal{U}_t(\boldsymbol{\zeta}^{(\mathcal{S})}) - \mathcal{U}_t(\boldsymbol{\zeta}^{(\mathcal{S} \cup \mathcal{A})}) + \langle \nabla \mathcal{U}_t(\boldsymbol{\zeta}^{(\mathcal{S})}), \, \boldsymbol{\zeta}^{(\mathcal{S} \cup \mathcal{A})} - \boldsymbol{\zeta}^{(\mathcal{S})} \rangle. \tag{10}$$

Rearranging and upper-bounding the right-hand side by maximizing over all feasible $\mathbf{v}$ supported on $\mathcal{S} \cup \mathcal{A}$ (legitimate, since $\boldsymbol{\zeta}^{(\mathcal{S} \cup \mathcal{A})}$ is itself such a feasible point),

$$
\begin{aligned}
\mathcal{U}_t(\boldsymbol{\zeta}^{(\mathcal{S} \cup \mathcal{A})}) - \mathcal{U}_t(\boldsymbol{\zeta}^{(\mathcal{S})}) &\leq \langle \nabla \mathcal{U}_t(\boldsymbol{\zeta}^{(\mathcal{S})}), \, \boldsymbol{\zeta}^{(\mathcal{S} \cup \mathcal{A})} - \boldsymbol{\zeta}^{(\mathcal{S})} \rangle - \frac{c_{\bar{\kappa}}}{2} \|\boldsymbol{\zeta}^{(\mathcal{S} \cup \mathcal{A})} - \boldsymbol{\zeta}^{(\mathcal{S})}\|^2 \\
&\leq \max_{\mathbf{v}: \, \mathbf{v}_{(\mathcal{S} \cup \mathcal{A})^c} = \mathbf{0}, \, \mathbf{v} \geq \mathbf{0}} \langle \nabla \mathcal{U}_t(\boldsymbol{\zeta}^{(\mathcal{S})}), \, \mathbf{v} - \boldsymbol{\zeta}^{(\mathcal{S})} \rangle - \frac{c_{\bar{\kappa}}}{2} \|\mathbf{v} - \boldsymbol{\zeta}^{(\mathcal{S})}\|^2.
\end{aligned}
\tag{11}
$$

The objective is separable and concave; its maximizer over $\mathbf{v} \geq \mathbf{0}$ supported on $\mathcal{S} \cup \mathcal{A}$ is

$$
\mathbf{v}_{\mathcal{S} \cup \mathcal{A}} = \max\left\{ \frac{1}{c_{\bar{\kappa}}} \nabla \mathcal{U}_{t, \mathcal{S} \cup \mathcal{A}}(\boldsymbol{\zeta}^{(\mathcal{S})}) + \boldsymbol{\zeta}^{(\mathcal{S})}_{\mathcal{S} \cup \mathcal{A}}, \, \mathbf{0} \right\},
\tag{12}
$$

so that

$$
(\mathbf{v} - \boldsymbol{\zeta}^{(\mathcal{S})})_{\mathcal{S} \cup \mathcal{A}} = \max\left\{ \frac{1}{c_{\bar{\kappa}}} \nabla \mathcal{U}_{t, \mathcal{S} \cup \mathcal{A}}(\boldsymbol{\zeta}^{(\mathcal{S})}), \, -\boldsymbol{\zeta}^{(\mathcal{S})}_{\mathcal{S} \cup \mathcal{A}} \right\}.
\tag{13}
$$

The KKT conditions for $f_{\mathcal{M}}(\mathcal{S}) = \max_{\mathbf{w} \geq \mathbf{0}, \, \mathrm{supp}(\mathbf{w}) \subseteq \mathcal{S}} \mathcal{U}_t(\mathbf{w})$ at $\boldsymbol{\zeta}^{(\mathcal{S})}$ require, for all $j \in \mathcal{S}$,

$$
\begin{cases}
\zeta^{(\mathcal{S})}_j > 0 \implies \nabla \mathcal{U}_{t,j}(\boldsymbol{\zeta}^{(\mathcal{S})}) = 0, \\
\zeta^{(\mathcal{S})}_j = 0 \implies \nabla \mathcal{U}_{t,j}(\boldsymbol{\zeta}^{(\mathcal{S})}) \leq 0,
\end{cases}
\tag{14}
$$

and hence $(\mathbf{v} - \boldsymbol{\zeta}^{(\mathcal{S})})_j = 0$ for all $j \in \mathcal{S}$. For $j \in \mathcal{A}$ we have $\zeta^{(\mathcal{S})}_j = 0$ (as $\mathcal{A} \cap \mathcal{S} = \emptyset$), so $(\mathbf{v} - \boldsymbol{\zeta}^{(\mathcal{S})})_j = \max\left\{ \frac{1}{c_{\bar{\kappa}}} \nabla \mathcal{U}_{t,j}(\boldsymbol{\zeta}^{(\mathcal{S})}), \, 0 \right\}$. Writing $\nabla \mathcal{U}^+_{t, \mathcal{A}}(\boldsymbol{\zeta}^{(\mathcal{S})}) = \max\left\{ \nabla \mathcal{U}_{t, \mathcal{A}}(\boldsymbol{\zeta}^{(\mathcal{S})}), \, \mathbf{0} \right\}$ and substituting the maximizer,

$$
0 \leq \mathcal{U}_t(\boldsymbol{\zeta}^{(\mathcal{S} \cup \mathcal{A})}) - \mathcal{U}_t(\boldsymbol{\zeta}^{(\mathcal{S})}) \leq \frac{1}{2c_{\bar{\kappa}}} \|\nabla \mathcal{U}^+_{t, \mathcal{A}}(\boldsymbol{\zeta}^{(\mathcal{S})})\|^2,
\tag{15}
$$

where the left inequality is the monotonicity of $f_{\mathcal{M}}$ (Lemma 5.1).

**Lower bound on the numerator.** Fix $j \in \mathcal{A}$. If $\nabla \mathcal{U}_{t,j}(\boldsymbol{\zeta}^{(\mathcal{S})}) \leq 0$, then $f_{\mathcal{M}}(\mathcal{S} \cup \{j\}) = f_{\mathcal{M}}(\mathcal{S})$ – monotonicity (Lemma 5.1) gives "$\geq$", while the optimality of $\boldsymbol{\zeta}^{(\mathcal{S})}$ with $\nabla \mathcal{U}_{t,j} \leq 0$ gives "$\leq$" – so such coordinates contribute zero to both the numerator and to $\|\nabla \mathcal{U}^+_{t, \mathcal{A}}\|^2$, and we may restrict to $j$ with $\nabla \mathcal{U}_{t,j}(\boldsymbol{\zeta}^{(\mathcal{S})}) > 0$. Let $\mathbf{1}^{(\{j\})}$ be the $j$-th standard basis vector, and for $\alpha \geq 0$ set $\mathbf{y}^{(\{j\})} = \boldsymbol{\zeta}^{(\mathcal{S})} + \alpha \mathbf{1}^{(\{j\})}$, so that $(\boldsymbol{\zeta}^{(\mathcal{S})}, \mathbf{y}^{(\{j\})}) \in \widetilde{\Omega}$. Since $\mathbf{y}^{(\{j\})}$ is feasible for $f_{\mathcal{M}}(\mathcal{S} \cup \{j\})$, restricted smoothness (Section 2) on $\widetilde{\Omega}$ yields

$$
\mathcal{U}_t(\boldsymbol{\zeta}^{(\mathcal{S} \cup \{j\})}) - \mathcal{U}_t(\boldsymbol{\zeta}^{(\mathcal{S})}) \geq \mathcal{U}_t(\mathbf{y}^{(\{j\})}) - \mathcal{U}_t(\boldsymbol{\zeta}^{(\mathcal{S})}) \geq \langle \nabla \mathcal{U}_t(\boldsymbol{\zeta}^{(\mathcal{S})}), \, \alpha \mathbf{1}^{(\{j\})} \rangle - \frac{\widetilde{C}_1}{2} \alpha^2.
$$

Maximizing the right-hand side over $\alpha$ at $\alpha = \nabla \mathcal{U}_{t,j}(\boldsymbol{\zeta}^{(\mathcal{S})})/\widetilde{C}_1 \geq 0$ gives

$$
f_{\mathcal{M}}(\mathcal{S} \cup \{j\}) - f_{\mathcal{M}}(\mathcal{S}) \geq \frac{1}{2\widetilde{C}_1} (\nabla \mathcal{U}_{t,j}(\boldsymbol{\zeta}^{(\mathcal{S})}))^2,
$$

and summing over $j \in \mathcal{A}$,

$$
\sum_{j \in \mathcal{A}} [f_{\mathcal{M}}(\mathcal{S} \cup \{j\}) - f_{\mathcal{M}}(\mathcal{S})] \geq \frac{1}{2\widetilde{C}_1} \|\nabla \mathcal{U}^+_{t, \mathcal{A}}(\boldsymbol{\zeta}^{(\mathcal{S})})\|^2.
\tag{16}
$$

**Combining.** Dividing (16) by (15), the common factor $\|\nabla \mathcal{U}^+_{t, \mathcal{A}}(\boldsymbol{\zeta}^{(\mathcal{S})})\|^2$ cancels and

$$
\gamma_{\mathcal{S}, \mathcal{A}} \geq \frac{c_{\bar{\kappa}}}{\widetilde{C}_1}.
$$

For $|\mathcal{S}|, |\mathcal{A}| \leq \kappa$ we have $\bar{\kappa} \leq 2\kappa$, hence $\Omega_{\bar{\kappa}} \subseteq \Omega_{2\kappa}$ and $c_{\bar{\kappa}} \geq c_{2\kappa}$ (Section 2). Therefore

$$
\gamma = \min_{\mathcal{S}, \mathcal{A}} \gamma_{\mathcal{S}, \mathcal{A}} \geq \frac{c_{2\kappa}}{\widetilde{C}_1} > 0,
$$

where the strict positivity follows since $\mathbf{K}_{\boldsymbol{\theta}_t} \succ \mathbf{0}$ ensures $c_{2\kappa} > 0$ (Lemma 5.3). This proves the claim. $\qquad \square$

# D. Dataset Details

## D.1. Molecular Instruction Datasets

The Molecular Instructions dataset (Fang et al., 2023) is a large-scale biomolecular instruction dataset for large language models designed to enhance a model's performance in various realms of biomolecular studies. It consists of three key components: (i) *molecule-oriented instructions*, (ii) *protein-oriented instructions* and *biomolecular text instructions*.

**Molecular-oriented instructions** consists of 148.4K instructions across six tasks, including *description-guided molecule prediction*, *retrosynthesis*, *forward reaction prediction* and *reagent prediction*.

The dataset distribution across various categories is shown in Figure 6(a). A detailed example of this specific section of the dataset is given in D.1.1.

**Protein-oriented instructions** focus on predicting the the structure, function, and activity of proteins, and facilitate protein design based on textual directives. It consists of 505K instructions spanning five categories of tasks.

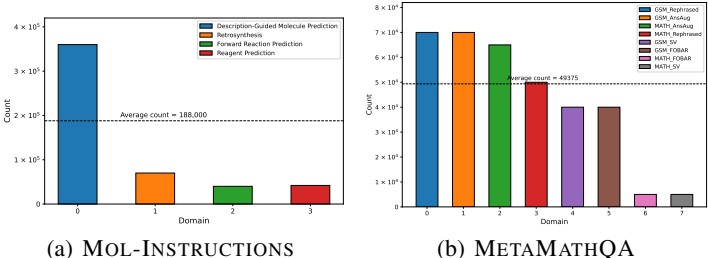

(a) MOL-INSTRUCTIONS      (b) METAMATHQA

*Figure 6.* Domain-wise sample distributions in MOL-INSTRUCTIONS and METAMATHQA, highlighting substantial domain imbalance in the number of instances contributed by different domains/tasks.

**Biomolecular text instructions** focus on bioinformatics and chemoinformatics, covering six information extraction and Q&A tasks represented through 53K instructions.

### D.1.1. DETAILED DESCRIPTION FOR MOLECULAR-ORIENTED INSTRUCTIONS

Table 2 shows the results obtained for the experiments carried out using the Mol Instructions datasets.

| Molecular Instruction Examples |
|---|

| Description-guided molecule design |
|---|

**Input:** Create a molecule with the structure as the one described. The molecule is a primary arylamine in which an amino functional group is substituted for one of the benzene hydrogens. It is a primary arylamine and a member of anilines.
**Output:**

```
[N][C][=C][C][=C][C][=C][Ring1][=Branch1]
```

| Forward reaction prediction |
|---|

**Input:** With the provided reactants and reagents, propose a potential product.

```
[O][=N+1][Branch1][C][O-1][C][=C][N][=C][Branch1][C][Cl][C][Branch1][C][I][=C][Ring1]↓
↪  [Branch2].[Fe]
```

**Output:**

```
[N][C][=C][N][=C][Branch1][C][Cl][C][Branch1][C][I][=C][Ring1][Branch2]
```

| Retrosynthesis |
|---|

**Input:** Please suggest potential reactants used in the synthesis of the provided product.

```
[C][=C][C][C][N][C][=Branch1][C][=O][O][C][Branch1][C][C][Branch1][C][C][C]
```

**Output:**

```
[C][=C][C][C][N].[C][C][Branch1][C][C][Branch1][C][C][O][C][=Branch1][C][=O][O][C][=B↓
↪  ranch1][C][=O][O][C][Branch1][C][C][Branch1][C][C][C]
```

---

**Reagent prediction**

**Input:** Please provide possible reagents based on the following chemical reaction.

```
[C][C][=C][C][=C][Branch1][C][N][C][=N][Ring1][#Branch1].[O][=C][Branch1][C][Cl][C][C↵
↪ l]>>[C][C][=C][C][=C][Branch1][Branch2][N][C][=Branch1][C][=O][C][Cl][C][=N][Ring↵
↪ 1][O]
```

**Output:**

```
[C][C][C][O][C][Ring1][Branch1].[C][C][N][Branch1][Ring1][C][C][C][C].[O]
```

---

## D.2. Reasoning Datasets

The METAMATHQA dataset (Yu et al., 2024a) is a large-scale mathematical question–answering dataset designed to train and evaluate large language models on mathematical reasoning. It comprises a wide range of mathematical domains, including algebra, calculus, and number theory. Each data instance contains a query (the mathematical problem) and a detailed step-by-step solution. The dataset distribution can be seen in Figure 6(b). In Table 1, we show results for experiments in which the model is trained on METAMATHQA and evaluated on various mathematical reasoning benchmarks described below.

## D.3. Evaluation Datasets

**NumGLUE** The NumGLUE dataset (Mishra et al., 2022) is a multi-task benchmark designed to evaluate the performance of AI systems on eight tasks that fundamentally require simple arithmetic understanding. The benchmark includes both newly created and previously existing arithmetic reasoning tasks, totaling approximately 100k questions. The tasks are intentionally imbalanced, for example, Task 1 contains around 400 examples, while Task 5 includes about 50k examples, reflecting real-world distributions in which arithmetic word problems are far more common than those requiring additional common-sense reasoning in addition to arithmetic reasoning.

**MATH** The MATH dataset (Hendrycks et al., 2021b) consists of 12,500 challenging competition-level mathematics problems. Each problem has a full step-by-step solution, enabling the training and evaluation of models that generate answer derivations and explanations.

The dataset spans seven subjects, namely Prealgebra, Algebra, Number Theory, Counting and Probability, Geometry, Intermediate Algebra, and Precalculus. Each subject comprises of problems from different difficulty level, ranging from level 1 to level 5.

**GSM8K** The GSM8K dataset (Cobbe et al., 2021) consists of 8.5K high-quality linguistically diverse grade-school math word problems which are designed to exhibit substantial linguistic variation while relying only on relatively simple grade-school mathematical concepts. The dataset is divided into 7.5K training examples and 1K test examples, with each problem typically requiring between 2 and 8 reasoning steps to solve.

**SVAMP** The SVAMP dataset (Patel et al., 2021) (Simple Variations on Arithmetic Math Word Problems) contains one-unknown arithmetic word problems up to approximately grade-4 level, designed by applying simple and controlled variations to problems in ASDiv (Miao et al., 2021) to highlight the brittleness of existing models when trained on standard math word-problem benchmarks.

SVAMP introduces several categories of variations, including: *(i) Question Sensitivity* i.e. same object with different structure, different object with same structure, or different object with a different structure; *(ii) Reasoning Ability* i.e. modifying relevant information, changing information, or inverting operations; and *(iii) Structural Invariance* i.e. adding irrelevant information, changing the order of objects, or changing the order of phrases.

**SimulEq** SimulEq is a subset of MAWPS (Koncel-Kedziorski et al., 2016), an online repository of math word problems designed to support research in automated math word-problem solving. SimulEq contains the highest-complexity equations among all the MAWPS subsets, including single equation and multi equation word problems.

**MMLU-Math** For our experiments, we use the mathematics-related subsets of the MMLU benchmark (Hendrycks et al., 2021a). We focus on four tasks that fall under the Elementary Mathematics category: *formal logic*, *high school mathematics*, *abstract algebra*, and *college mathematics*.

**AQuA** The AQuA dataset (Ling et al., 2017) consists of 100K multiple-choice math word problems in which each question is decomposed into four parts: the problem description (the *question*), the set of multiple-choice answer options (the *options*), the step-by-step explanation used to reach the correct answer (the *rationale*), and the correct option label.

# E. Implementation Details

## E.1. Hardware and License

All models are implemented in `Python 3.10` using `JAX`. All language model training were performed on servers with NVIDIA A100 GPUs.

## E.2. Finetuning experiments

We use JAX (Bradbury et al., 2018) for all our finetuning experiments. We use adam (Kingma & Ba, 2015) with learning rate of 2e-5, with $\hat{S}_t = 128$. We directly use raw lora gradients for constructing similarity matrices for CoLM, GREATS, PARTITIONSEL. For GREATS, ID and PARTITIONSEL we randomly sample 2-random points from val-set as anchors at every train step.

## E.3. Memory Efficient Gradient Computation

Since computing per-example gradients is prohibitively expensive, we rely on the last layer gradients of the LoRA params for computing the $\bar{\boldsymbol{\mu}}_{\boldsymbol{\theta}_t}$ and $\bar{\mathbf{K}}_{\boldsymbol{\theta}_t}$. Moreover, inspired by (Nguyen et al., 2025), we apply a adam-like lowpass filter on the raw gradients directly instead of their SPSA (Spall, 1992) proxy.

Concretely, let $\tilde{\boldsymbol{g}}_{\boldsymbol{\theta}_t}(\mathbf{z}_i)$ denote the raw last-layer LoRA gradient for example $\mathbf{z}_i$ at step $t$. We maintain exponentially smoothed first and second moments (bias-terms omitted for brevity)

$$\boldsymbol{m}_t(\mathbf{z}_i) = \beta_1 \, \boldsymbol{m}_{t-1}(\mathbf{z}_i) + (1 - \beta_1) \, \tilde{\boldsymbol{g}}_{\boldsymbol{\theta}_t}(\mathbf{z}_i),$$
$$\boldsymbol{v}_t(\mathbf{z}_i) = \beta_2 \, \boldsymbol{v}_{t-1}(\mathbf{z}_i) + (1 - \beta_2) \, \tilde{\boldsymbol{g}}_{\boldsymbol{\theta}_t}^2(\mathbf{z}_i).$$

We then define the filtered gradient feature used in the similarity matrices as $\boldsymbol{g}_{\boldsymbol{\theta}_t}(\mathbf{z}_i) = \frac{\boldsymbol{m}_t(\mathbf{z}_i)}{\sqrt{\boldsymbol{v}_t(\mathbf{z}_i) + \varepsilon}}$. Finally, we apply a randomized FFT projection (Ailon & Chazelle, 2006) to obtain a compressed features for constructing $\bar{\boldsymbol{\mu}}_{\boldsymbol{\theta}_t}$ and $\bar{\mathbf{K}}_{\boldsymbol{\theta}_t}$, which reduces memory movement to improve efficiency.

## E.4. Details of baselines

**GREATS (Wang et al., 2024).** GREATS formulates online batch selection as optimizing a set utility that measures the single-step reduction in validation loss under a gradient-descent update. Let $\boldsymbol{\theta}_t$ be the current parameters, $\mathcal{B}_t$ a candidate batch, and $\mathcal{S} \subseteq \mathcal{B}_t$ a subset of size $\kappa$. The ideal utility at iteration $t$ is

$$\mathcal{U}_t(\mathcal{S}) := \ell(\mathbf{z}_{\mathsf{val}}; \boldsymbol{\theta}_t) - \ell\left(\mathbf{z}_{\mathsf{val}}; \tilde{\boldsymbol{\theta}}_{t+1}(\mathcal{S})\right) \tag{17}$$

and selection solves $\arg\max_{\mathcal{S} \subseteq \mathcal{B}_t, \, |\mathcal{S}| = \kappa} \mathcal{U}_t(\mathcal{S})$. Since exact evaluation is intractable, GREATS applies a Taylor approximation to obtain a closed-form surrogate for the marginal gain of adding a training point $\mathbf{z}_i$:

$$\boldsymbol{\Delta}\mathcal{U}_t(\mathbf{z}_i \mid \mathcal{S}) = \ell(\mathbf{z}_{\mathsf{val}}; \tilde{\boldsymbol{\theta}}_{t+1}(\mathcal{S})) - \ell(\mathbf{z}_{\mathsf{val}}; \tilde{\boldsymbol{\theta}}_{t+1}(\mathcal{S} \cup \{\mathbf{z}_i\}))$$
$$\approx \eta_t \left\langle \boldsymbol{g}_{\boldsymbol{\theta}_t}(\mathbf{z}_i), \boldsymbol{g}_{\boldsymbol{\theta}_t}(\mathbf{z}_{\mathsf{val}}) \right\rangle - \eta_t^2 \left\langle \boldsymbol{g}_{\boldsymbol{\theta}_t}(\mathbf{z}_i), \boldsymbol{H}_{\mathbf{z}_{\mathsf{val}}}(\boldsymbol{\theta}_t)(\textstyle\sum_{\mathbf{z} \in \mathcal{S}} \boldsymbol{g}_{\boldsymbol{\theta}_t}(\mathbf{z})) \right\rangle.$$

In practice, GREATS further approximates $\boldsymbol{H}_{\mathbf{z}_{\mathsf{val}}}(\boldsymbol{\theta}_t) \approx \mathbf{I}$, yielding a gradient inner-product scoring with a correction term. A greedy procedure iteratively adds the point with largest approximate marginal gain until $k$ points are selected. To avoid materializing per-example model-sized gradients, GREATS computes all required gradient inner-products in a single backpropagation via a "ghost inner-product" reparameterization that expresses layerwise gradient inner-products using already-available activations and output gradients, and merges selection with the update without extra passes.

**CoLM** (Nguyen et al., 2025). CoLM casts mini-batch construction as coreset selection in gradient space for memory-efficient fine-tuning. CoLM first addresses imbalance by including *all* examples from "small" sources (those with insufficient sample count in the large batch), while selecting representatives (medoids) from each "big" source. To align selection with Adam, per-example gradients are normalized by the optimizer's exponential-moving-average statistics, yielding normalized directions proportional to $\boldsymbol{m}_t/(\epsilon + \sqrt{\boldsymbol{v}_t})$. To reduce dimensionality and denoise, CoLM estimates the gradient of the *last $V$-projection* parameters (e.g., LoRA $V$) using a zeroth-order SPSA estimator with two perturbed forward passes and precached penultimate activations, then sparsifies by keeping the coordinates with largest normalized magnitudes. Within each big source, a greedy medoid selection is performed in the projected, sparsified, Adam-normalized gradient space so that the aggregated coreset gradient approximates that of the full large batch; the final mini-batch is the union of all small-source examples and the selected big-source medoids.

**GradNorm** (Katharopoulos & Fleuret, 2018) Given a large batch $\mathcal{B}$, compute per-example gradient features and rank by norm. For parameters $\theta$ and loss $\ell_i = \ell(f_{\boldsymbol{\theta}}(\boldsymbol{x}_i), y_i)$, define raw gradient $\boldsymbol{g}_i = \nabla_{\boldsymbol{\theta}} \ell_i$. To align with Adam, each coordinate is normalized as $\tilde{\boldsymbol{g}}_i = \frac{\boldsymbol{m}_t}{\sqrt{\boldsymbol{v}_t} + \epsilon} \odot \boldsymbol{g}_i$ where $(\boldsymbol{m}_t, \boldsymbol{v}_t)$ are the exponential moving averages of first and second moments. Instead of $\ell_2$ distance, GradNorm computes similarity in this normalized gradient space using cosine: $s_{ij} = \frac{\langle \tilde{\boldsymbol{g}}_i, \tilde{\boldsymbol{g}}_j \rangle}{\|\tilde{\boldsymbol{g}}_i\|_2 \|\tilde{\boldsymbol{g}}_j\|_2}$.

Each example is scored by its (smoothed) gradient norm $\|\tilde{\boldsymbol{g}}_i\|_2$, and the top-$k$ are selected. This yields a subset whose update direction emphasizes examples with largest effective gradient magnitude under the optimizer's scaling.

### E.5. Training Details

Following the configuration in Yue et al. (2024), we employ a learning rate of $2 \times 10^{-5}$ with a cosine decay scheduler. The learning rate is linearly warmed up from 0 to $2 \times 10^{-5}$ during the first 3% of training steps and subsequently decays to 0 following a cosine schedule. We fix the maximum sequence length to 512 tokens. Unless otherwise stated, all experiments on METAMATHQA are trained for the equivalent of 2K gradient update steps. For randomised FFT projections, we project the gradients to respective models output embedding dimension.

For parameter-efficient fine-tuning, we adopt LoRA (Hu et al., 2022) with rank 64, scaling parameter $\alpha = 128$, and a dropout rate of 0.05. All experiments are conducted on A100 GPUs, and each configuration is repeated three times to account for variance in training.

## F. Additional Algorithms

This section provides implementation details for the inner optimization routine used by our selection objective. Recall from the main paper (Section 6) that the OMP subroutine (Algorithm 2) repeatedly solves a constrained concave maximization over weights supported on the current candidate set. Concretely, for a fixed support $S$, we optimize the utility $\mathcal{U}_t(\mathbf{w})$ over nonnegative weights $\mathbf{w} \geq \mathbf{0}$ with $\mathrm{supp}(\mathbf{w}) \subseteq S$.

Algorithm 3 describes the accelerated projected gradient ascent (APGA) solver (Manupriya et al., 2024b) we use for this inner problem. It is a standard Nesterov-style accelerated method with a projection step onto the nonnegative orthant, using a step size $1/\mathcal{L}$ where $\mathcal{L}$ is any Lipschitz constant of $\nabla \mathcal{U}_t$ (see Lemma 5.2 in the main paper). In our implementation, we warm-start APGA from the previous iterate whenever possible, and terminate when the relative improvement in objective becomes small (or after a fixed iteration budget), which keeps the overall overhead negligible.

---

**Algorithm 3** APGA: Accelerated Projected Gradient Ascent for the inner program (7)

---

1: **Input:** objective $\mathcal{U}_t(\mathbf{w})$; Lipschitz constant $\mathcal{L}$; initial point $\mathbf{w}_0 \in \mathbb{R}_+^{|\mathcal{B}_t|}$.
2: Initialize $y \leftarrow \mathbf{w}_0, \mathbf{w} \leftarrow \mathbf{w}_0, t \leftarrow 1$.
3: **while** not converged **do**
4:    $\mathbf{w}^+ \leftarrow \mathrm{PROJ}_{\geq 0}\left(y + \frac{1}{\mathcal{L}} \nabla \mathcal{U}_t(y)\right)$.
5:    $t^+ \leftarrow \frac{1 + \sqrt{1 + 4t^2}}{2}$.
6:    $y \leftarrow \mathbf{w}^+ + \frac{t-1}{t^+}(\mathbf{w}^+ - \mathbf{w})$.
7:    $(\mathbf{w}, t) \leftarrow (\mathbf{w}^+, t^+)$.
8: **end while**
9: **return** $\mathbf{w}$.

---

# G. Additional Experiments

## G.1. End-to-end wall-clock time and peak GPU memory

Here, we provide an end-to-end wall-clock time details for each baseline used in our experiments.

**Experimental setup.** To showcase the end-to-end wall-clock time for PARTITIONSEL as compared to other baselines, we train a QWEN2.5-3B model using a batch size of 16 with a selection fraction $\kappa = 0.25$ for 1K training steps.

| Method | Time (mins) |
|---|---|
| PartitionSel (Ours) | **51:08** |
| GREATS | 50.67 |
| COLM | 47:29 |
| ID | 53:25 |
| Random | 44.17 |

*Table 5.* End-to-end wall-clock time (minutes)

| Method | Memory |
|---|---|
| PartitionSel (Ours) | **31.5 GB** |
| ID | 32.9 GB |
| GREATS | 32.8 GB |
| COLM | 32.7 GB |
| Random | 28.0 GB |

*Table 6.* Peak GPU memory consumption.

**Efficiency-Accuracy Tradeoff**. Overall, PARTITIONSEL incurs comparable end-to-end training time relative to existing approaches, while achieving better accuracy and log-perplexity trade-offs as shown in our experiment analysis, demonstrating that its computational overhead is negligible in practice.

**Peak GPU memeory Cost**. We additionally report the peak GPU memory cost incurred by PARTITIONSEL against other baselines in Table 6. We observe that the peak GPU memory consumption of PARTITIONSEL is similar to existing gradient-based baselines. Random consumes the least peak memory as it does not perform subset selection and incurs no gradient computation.

## G.2. Handling extreme imbalance under proportional constraints

In our current data mixture experiments, we employ the METAMATHQA dataset for training, which is imbalanced. Figure 6(b) showcases the imbalance across different domains, where the minority domains are MATH_FOBAR and MATH_SV. In Figure 7, we provide the percentage of samples per domain (METAMATHQA).

In Figure 4 , we showcase the domain weights assigned by PARTITIONSEL where the minority domain weights are within a similar range as those assigned by DoReMi, thereby showing that the domain weight scheme of PARTITIONSEL (line 311) is robust to domain imbalance. We provide the log-perplexity curves in Figure , where PARTITIONSEL has better log-pplx reduction than DoReMi by 12.5% and 9.4% on minority (MATH_FOBAR, MATH_SV) domains, respectively. For completeness, Figure 8 shows representative values from the log-pplx plot.

## G.3. Pretraining and validation-set assumptions

It is unclear if the assumptions (e.g., access to a clean validation set) hold or scale effectively to the pretraining phase, where data mixtures are significantly larger and more diverse.

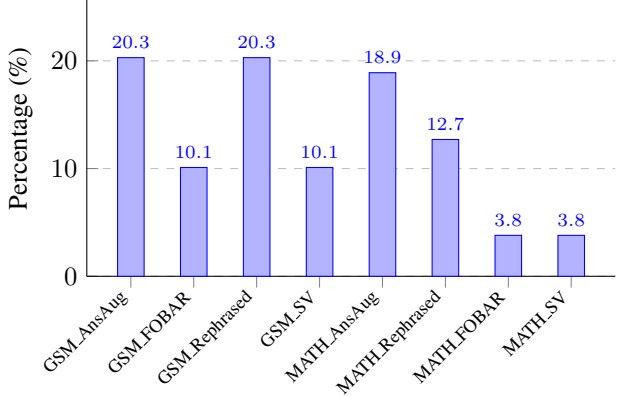

*Figure 7.* METAMATHQA domain composition.

While our primary empirical validation is in the supervised fine-tuning setting, we follow Wang et al. (2024) to implement pretraining as well. Since pretraining with DoReMI is a computational constraint for us (it requires additional overheads such as pretraining an additional proxy model before the main model pretraining begins), we compare against the independent coreset selection baseline ID.

*Table 7.* Pretraining validation log-pplx over training steps.

| Step | PARTITIONSEL | ID |
|------|------|------|
| 5K | 8.72 | 8.80 |
| 10K | 7.58 | 7.77 |
| 20K | 7.43 | 7.63 |
| 35K | 7.35 | 7.58 |
| 50K | 7.32 | 7.55 |

We train a LLAMA-3.2-1B model on 1.5B to-kens from a multi-domain corpus (DOLMA (Soldaini et al., 2024)) spanning heterogeneous sources (Common Crawl, Wikipedia, StackExchange, Reddit; $\sim$2.9M chunks total). We use a batch size of 16k tokens (32 sequences of length 512), learning rate of $3 \times 10^{-4}$ with linear decay, warmup of 1000 steps, and AdamW. In Table 7, we observe that our approach achieves better log-pplx reduction than domain-wise independent selection by 4.53%.

### G.4. Additional Results on LLAMA-3.1-8B

Table 9 reports the performance of LLAMA3.1-8B on MetamathQA (Yu et al., 2024a). We train the model for 2048 steps, with $\hat{S}_t$ as 128 and $\kappa$ as 16, thus admitting a subset selection ratio of 12.5%. 8 reports the performance on molecule generation (Fang et al., 2023), trained for 1024 steps.

*Figure 8.* Minority-domain validation log-pplx over training steps.

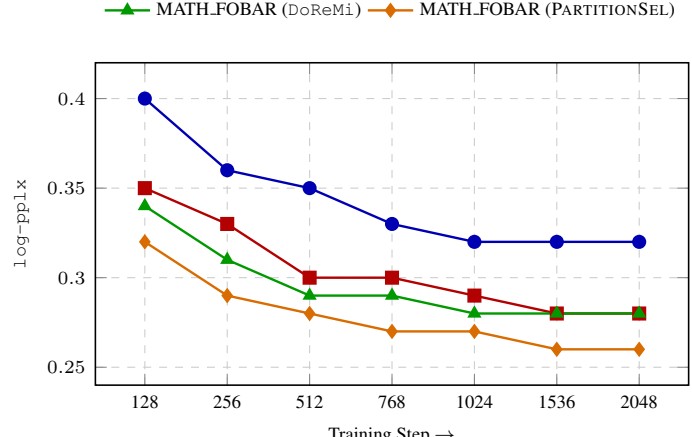

*Table 8.* Subset selection performance on four molecule generation datasets using BLEU / Edit Distance, evaluated with LLAMA-3.2-3B.

| Baseline | Desc. guided design | Forward reaction pred. | Reagent pred. | Retrosynthesis | Average |
|------|------|------|------|------|------|
| Random | 0.23 / 56.10 | 0.79 / 20.54 | 0.31 / 42.54 | 0.80 / 24.18 | 0.53 / 35.84 |
| COLM | 0.36 / 49.83 | 0.83 / 20.41 | 0.29 / 31.46 | 0.80 / **23.67** | 0.57 / 31.34 |
| GradNorm | 0.31 / 47.83 | 0.65 / 22.62 | 0.28 / 46.28 | 0.73 / 30.26 | 0.49 / 36.75 |
| GREATS | 0.33 / 51.80 | 0.83 / 20.51 | **0.50** / 25.44 | **0.82** / 24.11 | **0.62** / 30.46 |
| ID | 0.38 / 50.54 | 0.83 / 20.66 | 0.35 / 28.05 | 0.81 / 24.24 | 0.59 / 30.87 |
| IWD | 0.37 / 48.36 | **0.83** / 20.31 | 0.29 / 26.45 | 0.81 / 23.95 | 0.57 / **29.77** |
| PARTITIONSEL | **0.40** / **46.64** | 0.82 / **20.23** | 0.44 / 33.89 | 0.81 / 24.17 | **0.62** / 31.23 |

### G.5. Additional Results on LLAMA-3.2-3B

**Setup.** We report results for LLAMA-3.2-3B fine-tuned on METAMATHQA (Yu et al., 2024a) for 2048 optimization steps using a full batch size $|\hat{S}_t| = 128$ and a selection budget $\kappa = 16$, corresponding to a 12.5% subset ratio. We evaluate the resulting model on eight math-reasoning benchmarks and report accuracy (%) in Table 10.

**Results and discussion.** Across the suite of benchmarks, PARTITIONSEL achieves the best average performance (45.32) among the baselines reported, while remaining competitive on individual tasks. We observe consistent gains on several compositional and multi-step reasoning benchmarks (e.g., NumGLUE and SimulEq), suggesting that coordinating subset selection across domains helps preserve a broader set of skills compared to domain-wise selection strategies. Moreover, relative to COLM and GradNorm, the improvements are achieved without requiring a separate domain-weight learning stage, aligning with our goal of a simple batch-level selection mechanism.

| Method | NumGLUE | MMLU-Math | GSM8K | SVAMP | SimulEq | DeepMind | AQuA | SAT | Avg |
|---|---|---|---|---|---|---|---|---|---|
| COLM (Nguyen et al., 2025) | 38.00 | **40.35** | 48.22 | **60.80** | 17.12 | **16.60** | 31.50 | 39.09 | 36.46 |
| GradNorm (Katharopoulos & Fleuret, 2018) | 36.85 | 39.53 | 44.81 | 54.10 | 11.28 | 15.80 | 31.50 | 40.91 | 34.35 |
| ID | 41.27 | 36.65 | 48.90 | 58.40 | 18.87 | 15.80 | 34.25 | 38.64 | 36.60 |
| PARTITIONSEL | **43.95** | 39.01 | **49.13** | 56.70 | **19.65** | 15.90 | **36.61** | **44.09** | **38.13** |

*Table 9.* Downstream Accuracy (%) on evaluation datasets for LLAMA-3.1-8B trained on METAMATHQA with a 12.5% subset ratio.

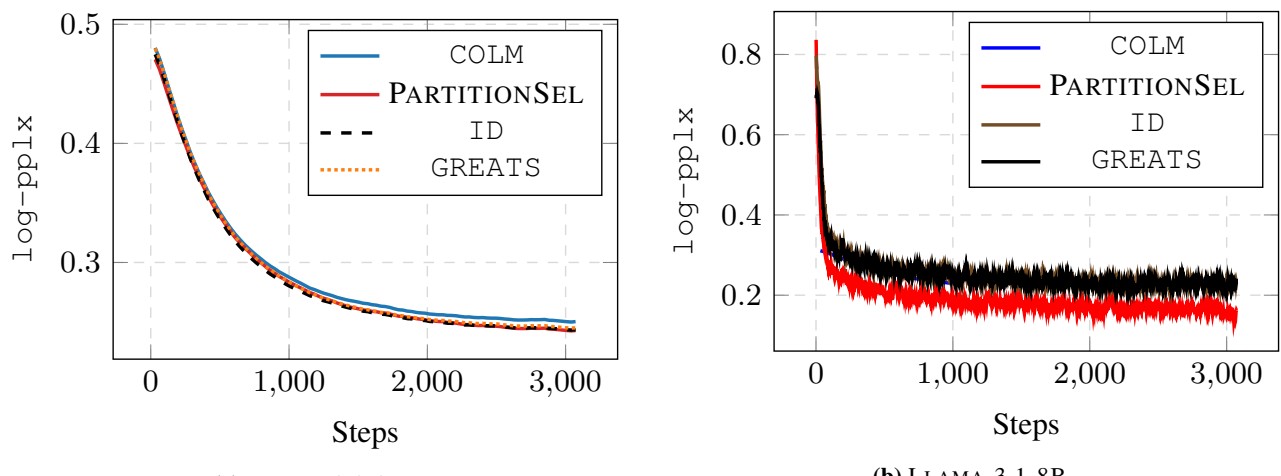

**(a)** LLAMA-3.2-3B  **(b)** LLAMA-3.1-8B

*Figure 9.* Validation log-pplx on METAMATHQA.

| Method | NumGLUE | MMLU-Math | GSM8K | SVAMP | SimulEq | DeepMind | AQuA | SAT | Avg |
|---|---|---|---|---|---|---|---|---|---|
| COLM (Nguyen et al., 2025) | 47.02 | 37.68 | 58.07 | **71.70** | 24.51 | 23.30 | **46.06** | 46.36 | 44.34 |
| GradNorm (Katharopoulos & Fleuret, 2018) | 47.60 | **45.99** | 59.21 | 71.00 | 22.96 | 22.60 | 40.55 | 45.91 | 44.48 |
| GREATS (Wang et al., 2024) | 49.81 | 32.65 | **60.65** | 67.90 | 25.88 | 20.60 | 41.34 | **52.27** | 43.89 |
| PARTITIONSEL | **50.29** | 39.63 | 59.82 | 69.90 | **30.16** | **24.00** | 37.40 | 51.36 | **45.32** |

*Table 10.* Downstream Accuracy (%) on evaluation datasets for LLAMA-3.2-3B trained on METAMATHQA with a 12.5% subset ratio.

### G.6. Additional Results on QWEN2.5-7B

**Setup.**   We report results for QWEN2.5-7B fine-tuned on METAMATHQA (Yu et al., 2024a) for 2048 steps with full batch size $|\hat{S}_t| = 128$ and a selection budget $\kappa = 16$ (subset ratio 12.5%). We evaluate accuracy (%) on the same set of eight math-reasoning benchmarks used in the main paper; results are summarized in Table 11.

| Method | NumGLUE | MMLU-Math | GSM8K | SVAMP | SimulEq | DeepMind | AQuA | SAT | Avg |
|---|---|---|---|---|---|---|---|---|---|
| ID | **57.68** | **64.68** | 80.52 | 84.60 | **41.44** | 37.90 | 60.24 | 78.18 | 63.15 |
| PARTITIONSEL | 56.43 | 64.58 | **83.47** | **86.40** | 38.72 | **38.10** | **63.78** | **83.64** | **64.39** |

*Table 11.* Downstream Accuracy (%) on evaluation datasets for QWEN2.5-7B trained on METAMATHQA with a 12.5% subset ratio.

**Results and discussion.**   PARTITIONSEL improves the average accuracy over the strongest reported baseline (ID) from 63.15 to 64.39, with particularly notable gains on GSM8K, SVAMP, AQuA, and SAT. Interestingly, ID remains slightly better on NumGLUE and SimulEq, suggesting that while joint selection generally improves breadth, some benchmarks may benefit from more domain-specialized selections. Overall, the results reinforce that the proposed joint selection strategy scales to larger backbones (7B) and preserves the accuracy gains observed in the 3B setting.

## H. Code

Our Code is available at `https://github.com/efficiency-learning/PartitionSEL`

