## Supplementary Material: Minibatch selection for Language Models via Partition Matroid Constrained Gradient Matching

## A. Organization of Appendix

The Appendix is structured as follows. Section C summarizes the main theoretical results. Section F presents the detailed description of the proposed algorithm. Sections E and E.4 outline the implementation aspects and the specifics of the experimental setup, respectively. Finally, Section H provides a link to the publicly available codebase.

## B. Background

### B.1. Approximate Submodularity

We adopt the notion of *approximate submodularity* characterized by the *submodularity ratio* as shown in (Das & Kempe, 2018). For a monotone set function $f$, the submodularity ratio with respect to a ground set $\mathcal{V}$, a reference set $\mathcal{S} \subseteq \mathcal{V}$, and a parameter $\kappa \geq 1$ is defined as

$$\alpha_{\mathcal{V},\kappa}(f) = \min_{\substack{\mathcal{S} \subseteq \mathcal{V},\ \mathcal{A}:\, |\mathcal{A}| \leq \kappa, \\ \mathcal{A} \cap \mathcal{S} = \varnothing}} \frac{\sum_{u \in \mathcal{A}} (f(\mathcal{S} \cup \{u\}) - f(\mathcal{S}))}{f(\mathcal{S} \cup \mathcal{A}) - f(\mathcal{S})}, \qquad \text{with } 0/0 := 1.$$

The function $f(\cdot)$ is submodular if and only if $\alpha_{\mathcal{V},\kappa}(f) \geq 1$. More generally, if

$$\alpha \equiv \frac{\sum_{u \in \mathcal{A}} (f(\mathcal{S} \cup \{u\}) - f(\mathcal{S}))}{f(\mathcal{S} \cup \mathcal{A}) - f(\mathcal{S})} > 0,$$

but not necessarily $\alpha \geq 1$, then $f$ is said to be $\alpha$-*weakly submodular*.

## C. Theoretical Results

**Lemma C.1.** ($\mathcal{L}$-smoothness) *The function* $\mathcal{U}_t(\cdot) : \mathbb{R}^{|\mathcal{B}_t|} \to \mathbb{R}$ *is* $\mathcal{L}$-*smooth. That is, for all* $\mathbf{w}_1, \mathbf{w}_2 \in \mathbb{R}^{|\mathcal{B}_t|}$, $\|\nabla \mathcal{U}_t(\mathbf{w}_1) - \nabla \mathcal{U}_t(\mathbf{w}_2)\| \leq \mathcal{L} \|\mathbf{w}_1 - \mathbf{w}_2\|$.

**Lemma C.2.** (Finite RSC/RSM) *Let* $\mathbf{K}_{\boldsymbol{\theta}_t} \in \mathbb{R}^{|\mathcal{B}_t| \times |\mathcal{B}_t|}$ *be symmetric positive definite and consider the concave quadratic* $\mathcal{U}_t(\mathbf{w}) := \mathbf{w}^\top \boldsymbol{\mu}_{\boldsymbol{\theta}_t} - \frac{1}{2} \mathbf{w}^\top \mathbf{K}_{\boldsymbol{\theta}_t} \mathbf{w}$. *Then, for any sparsity level* $\kappa \in \mathbb{N}$, $\mathcal{U}_t(\cdot)$ *is restricted strongly concave and restricted smooth on the set of* $\kappa$-*sparse vectors, with parameters given by the* $\kappa$-*sparse extremal eigenvalues of* $\mathbf{K}_{\boldsymbol{\theta}_t}$.

**Lemma C.2.** (Finite RSC/RSM) *Let* $\mathbf{K}_{\boldsymbol{\theta}_t} \in \mathbb{R}^{|\mathcal{B}_t| \times |\mathcal{B}_t|}$ *be symmetric positive definite and consider the concave quadratic* $\mathcal{U}_t(\mathbf{w}) := \mathbf{w}^\top \boldsymbol{\mu}_{\boldsymbol{\theta}_t} - \frac{1}{2} \mathbf{w}^\top \mathbf{K}_{\boldsymbol{\theta}_t} \mathbf{w}$. *Then, for any sparsity level* $\kappa \in \mathbb{N}$, $\mathcal{U}_t(\cdot)$ *is restricted strongly concave and restricted smooth on the set of* $\kappa$-*sparse vectors, with parameters given by the* $\kappa$-*sparse extremal eigenvalues of* $\mathbf{K}_{\boldsymbol{\theta}_t}$.

*Proof.* The function $\mathcal{U}_t(\mathbf{w}) = \mathbf{w}^\top \boldsymbol{\mu}_{\boldsymbol{\theta}_t} - \frac{1}{2} \mathbf{w}^\top \mathbf{K}_{\boldsymbol{\theta}_t} \mathbf{w}$ is concave. We calculate $\mathcal{U}_t(\mathbf{w}_1) - \mathcal{U}_t(\mathbf{w}_2) - \nabla \langle \mathcal{U}_t(\mathbf{w}_2), \mathbf{w}_1 - \mathbf{w}_2 \rangle = -0.5(\mathbf{w}_1 - \mathbf{w}_2)^T \mathbf{K}_{\boldsymbol{\theta}_t}(\mathbf{w}_1 - \mathbf{w}_2)$. Since $\mathbf{w}$ is $k_1 + k_2$ sparse, we consider $\mathbf{w}_1$ as $k_1^1 + k_2^1$ and $\mathbf{w}_2$ as $k_1^2 + k_2^2$ sparse vectors respectively. Now, $\Delta \mathbf{w} = \mathbf{w}_1 - \mathbf{w}_2$ has a maximum of $k \leq k_1^1 + k_2^1 + k_1^2 + k_2^2$ non-zero entries. For the constants $b$ and $B$ satisfying $-b\|\Delta\mathbf{w}\|^2 \geq -\Delta\mathbf{w}^T \mathbf{K}_{\boldsymbol{\theta}_t} \Delta\mathbf{w} \geq -B\|\Delta\mathbf{w}\|^2$, we have $b \geq k$-sparse smallest eigenvalue of $\mathbf{K}_{\boldsymbol{\theta}_t}$ and $B \leq k$-sparse largest eigenvalue of $\mathbf{K}_{\boldsymbol{\theta}_t}$. $\square$

**Lemma 5.1.** (Monotonicity) *Let* $f_{\mathcal{M}}(\cdot)$ *be defined as in* (10). *Then for any index sets* $\tilde{A} \subseteq \tilde{B}$ *we have* $f_{\mathcal{M}}(\tilde{A}) \leq f_{\mathcal{M}}(\tilde{B})$.

*Proof.* Let $|\tilde{A}| = n_1$ and $|\tilde{B}| = n_2$ and since $\tilde{A} \subseteq \tilde{B}$ we have $n_1 < n_2$. We index the elements in $\tilde{B}$ such that the first $n_1$ elements are contained in $\tilde{A}$.

$$f_{\mathcal{M}}(\tilde{B}) = \max_{\mathbf{w} \in \mathbb{R}_+^{n_2},\, supp(\mathbf{w}) \subseteq \tilde{B}} \mathcal{U}_t(\mathbf{w}) \geq \max_{\mathbf{w} \in \mathbb{R}_+^{n_1},\, supp(\mathbf{w}) \subseteq \tilde{A}} \mathcal{U}_t(\mathbf{w}) = f_{\mathcal{M}}(\tilde{A}) \tag{14}$$

□

**Theorem C.3.** *(Weak Submodularity) The set function $f_{\mathcal{M}}(\cdot)$ in (10) is weakly submodular with the submodularity ratio $\alpha > 0$*

*Proof.* As per definition of weakly-submodularity in order to get towards minimum submodularity ratio, we lower bound the numerator and upper bound the denominator. Let $\bar{\kappa} = |\mathcal{S}| + |\mathcal{A}|$. Recall that $\zeta^{(\mathcal{S})}, \zeta^{(\mathcal{S} \cup \mathcal{A})} \in \mathbb{R}^d$ are the maximizers $\mathcal{U}_t(\zeta^{(\mathcal{S})}) = f(\mathcal{S})$ and $\mathcal{U}_t(\zeta^{(\mathcal{S} \cup \mathcal{A})}) = f(\mathcal{S} \cup \mathcal{A})$ respectively.

By definition of *RSC* and *RSM* constants (from 2) we find

$$\frac{c_{\bar{\kappa}}}{2} \|\zeta^{(\mathcal{S} \cup \mathcal{A})} - \zeta^{(\mathcal{S})}\|^2 \leq \mathcal{U}_t(\zeta^{(\mathcal{S})}) - \mathcal{U}_t(\zeta^{(\mathcal{S} \cup \mathcal{A})}) + \left\langle \nabla \mathcal{U}_t(\zeta^{(\mathcal{S})}), \zeta^{(\mathcal{S} \cup \mathcal{A})} - \zeta^{(\mathcal{S})} \right\rangle. \tag{15}$$

Since via Lemma 5.1 $f_{\mathcal{M}}(\cdot)$ is monotone for increasing supports we get

$$\mathcal{U}_t(\zeta^{(\mathcal{S} \cup \mathcal{A})}) - \mathcal{U}_t(\zeta^{(\mathcal{S})}) \leq \left\langle \nabla \mathcal{U}_t(\zeta^{(\mathcal{S})}), \zeta^{(\mathcal{S} \cup \mathcal{A})} - \zeta^{(\mathcal{S})} \right\rangle - \frac{c_{\bar{\kappa}}}{2} \|\zeta^{(\mathcal{S} \cup \mathcal{A})} - \zeta^{(\mathcal{S})}\|^2$$
$$\leq \max_{\mathbf{v}: \mathbf{v}_{(\mathcal{S} \cup \mathcal{A})^c} = 0; \mathbf{v} > \mathbf{0}} \left\langle \nabla \mathcal{U}_t(\zeta^{(\mathcal{S})}), \mathbf{v} - \zeta^{(\mathcal{S})} \right\rangle - \frac{c_{\bar{\kappa}}}{2} \|\mathbf{v} - \zeta^{(\mathcal{S})}\|^2 \tag{16}$$

The vector $\mathbf{v}$ with support restricted to the coordinates by $\mathcal{S} \cup \mathcal{A}$ attains maximum at

$$\mathbf{v}_{\mathcal{S} \cup \mathcal{A}} = \max\left\{ \frac{1}{c_{\bar{\kappa}}} \nabla \mathcal{U}_{t, \mathcal{S} \cup \mathcal{A}}(\zeta^{(\mathcal{S})}) + \zeta^{(\mathcal{S})}_{\mathcal{S} \cup \mathcal{A}}, \mathbf{0} \right\} \tag{17}$$

It then follows that

$$(\mathbf{v} - \zeta^{(\mathcal{S})})_{\mathcal{S} \cup \mathcal{A}} = \max\left\{ \frac{1}{c_{\bar{\kappa}}} \nabla \mathcal{U}_{t, \mathcal{S} \cup \mathcal{A}}(\zeta^{(\mathcal{S})}), -\zeta^{(\mathcal{S})}_{\mathcal{S} \cup \mathcal{A}} \right\} \tag{18}$$

The KKT conditions at the optimum $\zeta^{(\mathcal{S})}$ for the function $f(\mathcal{S})$ necessitates that $\forall j \in \mathcal{S}$,

$$\begin{aligned} \zeta^{(\mathcal{S})}_j > 0 &\implies \nabla \mathcal{U}_{t,j}(\zeta^{(\mathcal{S})}) = 0, \\ \zeta^{(\mathcal{S})}_j = 0 &\implies \nabla \mathcal{U}_{t,j}(\zeta^{(\mathcal{S})}) \leq 0 \end{aligned} \tag{19}$$

and hence we have $(\mathbf{v} - \zeta^{(\mathcal{S})})_j = 0$

For $j \in \mathcal{A}$, $\zeta^{(\mathcal{A})}_j = 0$ which implies $(\mathbf{v} - \zeta^{(\mathcal{S})})_j = \max\left\{ \frac{1}{c_{\bar{\kappa}}} \nabla \mathcal{U}_{t;j}(\zeta^{(\mathcal{S})}), 0 \right\}$.

Defining $\nabla \mathcal{U}^+_{t,\mathcal{A}}(\zeta^{(\mathcal{S})}) = \max\{\nabla \mathcal{U}^+_{t,\mathcal{A}}(\zeta^{(\mathcal{S})}), \mathbf{0}\}$ and plugging the quantities computed at the maximum value $\mathbf{v}$ in we get the bound

$$0 \leq \mathcal{U}_t(\zeta^{(\mathcal{S} \cup \mathcal{A})}) - \mathcal{U}_t(\zeta^{(\mathcal{S})}) \leq \frac{1}{2c_{\bar{\kappa}}} \|\nabla l^+_{\mathcal{A}}(\zeta^{(\mathcal{S})})\|^2. \tag{20}$$

In order to lower bound the numerator, we consider a single coordinate $j \in \mathcal{A}$. It suffices to restrict to those coordinates $j$ where $\nabla \mathcal{U}_{t;j}(\zeta^{(\mathcal{S})}) > 0$.

Let $\mathbf{1}^{(\{j\})}$ be the vector with a value of 1 only at the $j$-th coordinates and zero elsewhere. For a $\alpha \geq 0$, we define $\mathbf{y}^{(\{j\})} = \zeta^{(\mathcal{S})} + \alpha \mathbf{1}^{\{j\}}$ such that $(\zeta^{(\mathcal{S})}, \mathbf{y}^{(\{j\})}) \in \Omega$. As $\zeta^{(\mathcal{S} \cup \{j\})}$ is the optimal point for $f_{\mathcal{M}}(\mathcal{S} \cup \{j\})$ we have

$$\mathcal{U}_t(\zeta^{(\mathcal{S} \cup \{j\})}) - \mathcal{U}_t(\zeta^{(\mathcal{S})}) \geq \mathcal{U}_t(\mathbf{y}^{(\{j\})}) - \mathcal{U}_t(\zeta^{(\mathcal{S})}) \geq \langle \nabla \mathcal{U}_t(\zeta^{(\mathcal{S})}), \alpha \mathbf{1}^{(\{j\})} \rangle - \frac{\widetilde{C}_1}{2} \alpha^2$$

Maximizing w.r.t $\alpha$ we get $\alpha = \frac{\nabla \mathcal{U}_{t;j}(\boldsymbol{\zeta}^{(\mathcal{S})})}{\widetilde{C}_1} \geq 0$. Substituting the maximum value we get

$$\mathcal{U}_t(\boldsymbol{\zeta}^{(\mathcal{S} \cup \{j\})}) - \mathcal{U}_t(\boldsymbol{\zeta}^{(\mathcal{S})}) \geq \frac{1}{2\widetilde{C}_1}\left(\nabla \mathcal{U}_{t;j}(\boldsymbol{\zeta}^{(\mathcal{S})})\right)^2$$

$$\implies \sum_{j \in \mathcal{A}}\left[\mathcal{U}_t(\boldsymbol{\zeta}^{(\mathcal{S} \cup \{j\})}) - \mathcal{U}_t(\boldsymbol{\zeta}^{(\mathcal{S})})\right] \geq \frac{1}{2\widetilde{C}_1}\|\nabla \mathcal{U}_{t,\mathcal{A}}^+(\boldsymbol{\zeta}^{(\mathcal{S})})\|^2$$

We obtain $\gamma_{\mathcal{S},\mathcal{A}} \geq \frac{c_{\bar{\kappa}}}{\widetilde{C}_1}$. Taking the minimum over all sets $\mathcal{S}$ and $\mathcal{A}$ proves the theorem.

$\square$

**Proposition C.1** (Positive semidefiniteness of $\mathbf{K}_{\boldsymbol{\theta}_t}$). *Let $\mathbf{K}_{\boldsymbol{\theta}_t}[i,j] = \eta^2\, \mathbf{g}_{\boldsymbol{\theta}_t}(\mathbf{z}_i)^\top \boldsymbol{H}_{\mathbf{z}_{\mathrm{val}}}\mathbf{g}_{\boldsymbol{\theta}_t}(\mathbf{z}_j)$ where $\boldsymbol{H}_{\mathbf{z}_{\mathrm{val}}} \succeq 0$. Then the matrix $\mathbf{K}_{\boldsymbol{\theta}_t}$ is positive semidefinite.*

*Proof.* We define
$$G_{\boldsymbol{\theta}_t} = \begin{pmatrix} \mathbf{g}_{\boldsymbol{\theta}_t}(\mathbf{z}_1) & \mathbf{g}_{\boldsymbol{\theta}_t}(\mathbf{z}_2) & \cdots & \mathbf{g}_{\boldsymbol{\theta}_t}(\mathbf{z}_n) \end{pmatrix} \in \mathbb{R}^{d \times n}.$$

Then
$$\mathbf{K}_{\boldsymbol{\theta}_t} = \eta^2\, G_{\boldsymbol{\theta}_t}^\top \boldsymbol{H}_{\mathbf{z}_{\mathrm{val}}}G_{\boldsymbol{\theta}_t}.$$

Consider the block matrix
$$M = \begin{pmatrix} \boldsymbol{H}_{\mathbf{z}_{\mathrm{val}}} & G_{\boldsymbol{\theta}_t} \\ G_{\boldsymbol{\theta}_t}^\top & \mathbf{K}_{\boldsymbol{\theta}_t}/\eta^2 \end{pmatrix}.$$

Since $\boldsymbol{H}_{\mathbf{z}_{\mathrm{val}}} \succeq 0$, assume $\boldsymbol{H}_{\mathbf{z}_{\mathrm{val}}} \succ 0$ (or restrict to its support). The Schur complement of $\boldsymbol{H}_{\mathbf{z}_{\mathrm{val}}}$ in $M$ is

$$\mathbf{K}_{\boldsymbol{\theta}_t}/\eta^2 - G_{\boldsymbol{\theta}_t}^\top \boldsymbol{H}_{\mathbf{z}_{\mathrm{val}}}^{-1}G_{\boldsymbol{\theta}_t}.$$

By the definition of $\mathbf{K}_{\boldsymbol{\theta}_t}$, this Schur complement is identically zero. Therefore,

$$M \succeq 0.$$

By the Schur complement lemma, this implies
$$\mathbf{K}_{\boldsymbol{\theta}_t}/\eta^2 \succeq 0,$$

and hence
$$\mathbf{K}_{\boldsymbol{\theta}_t} \succeq 0.$$

$\square$

**Theorem 5.3.** *If $\hat{S}_t$ is the $\kappa$-sparse subset obtained from PARTITIONSEL and $\mathcal{S}^*$ is the optimal $\kappa$-sparse subset then,*

$$f_{\mathcal{M}}(\hat{S}_t) \geq f(\mathcal{S}^*)(1 - e^{-\frac{c_{2\kappa}}{\widetilde{C}_1}}) \tag{11}$$

*Proof.* Let $\mathcal{S} = \hat{\mathcal{S}}_i$ be the set chosen by PARTITIONSEL upto the iteration $i$ such that $\hat{\mathcal{S}}_\kappa = \hat{\mathcal{S}}$ and $\mathcal{S}^*$ be the optimal set. We define the residual set $\mathcal{S}_R = \mathcal{S}^* \setminus \mathcal{S}$. Given $\mathcal{S}$, let $v$ be the index that would be selected by the running next step of PARTITIONSEL. Let $D(i+1) = f(\mathcal{S} \cup \{v\}) - f(\mathcal{S})$. Defining $\mathbf{y}^{(\{v\})} = \boldsymbol{\zeta}^{(\mathcal{S})} + \alpha \mathbf{1}^{(\{v\})}$ for some $\alpha \geq 0$.

Since $\boldsymbol{\zeta}^{(\mathcal{L} \cup \{v\})}$ is the maximizing point for $f(\mathcal{S} \cup \{v\})$ we get

$$D(i+1) \geq \mathcal{U}(\mathbf{y}^{(\{v\})}) - \mathcal{U}(\boldsymbol{\zeta}^{(\mathcal{S})}) \geq \langle \nabla \mathcal{U}(\boldsymbol{\zeta}^{(\mathcal{S})}), \alpha \mathbf{1}^{(\{v\})}\rangle - \frac{\widetilde{C}_1}{2}\alpha^2 \tag{21}$$

Setting $\alpha = \frac{\nabla \mathcal{U}_v^+(\boldsymbol{\zeta}^{(\mathcal{S})})}{\widetilde{C}_1}$ we get

$$D(i+1) \geq \frac{1}{2\tilde{C}_1}[\nabla \mathcal{U}_v^+(\boldsymbol{\zeta}^{(\mathcal{S})})]^2 \geq \frac{1}{2\kappa\tilde{C}_1} \sum_{j \in \mathcal{S}_R}[\nabla \mathcal{U}_j^+(\boldsymbol{\zeta}^{(S)})]^2 \tag{22}$$

where use the inequalities that $c_{\tilde{\kappa}} \geq c$ □

**Independent domain selection**: We also create the following independent domain selection baseline of (Wang et al., 2024) in which The budget for each domain is $\kappa_c^t$. The resulting objective is

$$f^{\mathsf{Ind}}(S) = \sum_{c=1}^{\mathcal{C}} \max_{\mathbf{w}_c \in \{0,1\}^{|\mathcal{B}_t^c|}, \|\mathbf{w}_c\|_0 \leq \kappa_c^t} \mathcal{U}_t(\mathbf{w}_c, \mathcal{B}_t^c; \mathbf{z}_{\mathsf{val}}) \tag{23}$$

The overall selection is required to satisfy the feasibility condition $\mathsf{supp}(\cup_{c=1}^{\mathcal{C}}\mathbf{w}_c) \subseteq \mathcal{S}$.

## D. Dataset Details

### D.1. Molecular Instruction Datasets

The Molecular Instructions dataset (Fang et al., 2023) is a large-scale biomolecular instruction dataset for large language models designed to enhance a model's performance in various realms of biomolecular studies. It consists of three key components: molecule-oriented instructions, protein-oriented instructions, and biomolecular text instructions.

**Molecular-oriented instructions** shed light on the fundamental challenges of diverse chemical reactions and molecular design. It consists of 148.4K instructions across six tasks, including description-guided molecule prediction, retrosynthesis, forward reaction prediction, and reagent prediction. The dataset distribution across various categories is shown in Figure 6. A detailed example of the dataset is given in D.1.

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

PARTITIONSEL. For GREATS, ID and PARTITIONSELwe randomly sample 2-random points from val-set as anchors at every train step.

### E.3. Details of baselines

**GREATS (Wang et al., 2024).**   GREATS formulates online batch selection as optimizing a set utility that measures the single-step reduction in validation loss under a gradient-descent update. Let $\boldsymbol{\theta}_t$ be the current parameters, $B_t$ a candidate batch, and $S \subseteq B_t$ a subset of size $k$. The ideal utility at iteration $t$ is

$$\mathcal{U}^{(t)}(S; \boldsymbol{z}^{(\text{val})}) := \ell(\boldsymbol{\theta}_t, \boldsymbol{z}^{(\text{val})}) - \ell\left(\boldsymbol{\theta}_t - \eta_t \sum_{\boldsymbol{z} \in S} \nabla\ell(\boldsymbol{\theta}_t, \boldsymbol{z}), \boldsymbol{z}^{(\text{val})}\right),$$

and selection solves $\arg\max_{S \subseteq B_t, |S|=k} \mathcal{U}^{(t)}(S; \boldsymbol{z}^{(\text{val})})$. Since exact evaluation is intractable, GREATS applies a lower-order Taylor approximation of the validation loss around $\boldsymbol{\theta}_t$ to obtain a closed-form surrogate for the marginal gain of adding a training point $\boldsymbol{z}$:

$$U^{(t)}(\boldsymbol{z} \mid S) \approx \eta_t \, \mathbf{g}(\boldsymbol{z})^\top \mathbf{g}(\boldsymbol{z}^{(\text{val})}) - \eta_t^2 \, \mathbf{g}(\boldsymbol{z})^\top H(\boldsymbol{z}^{(\text{val})}) \, \mathbf{g}(\boldsymbol{z}^*),$$

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

| Baseline | Desc. guided design | Forward reaction pred. | Reagent pred. | Retrosynthesis |
|---|---|---|---|---|
| Random | 0.23 / 56.10 | 0.79 / 20.54 | 0.31 / 42.54 | 0.80 / 24.18 |
| ID | 0.22 / 56.34 | 0.78 / 20.56 | 0.34 / 39.09 | 0.80 / 24.03 |
| PARTITIONSEL | 0.22 / 56.55 | 0.78 / 20.34 | 0.36 / 41.28 | 0.81 / 25.09 |

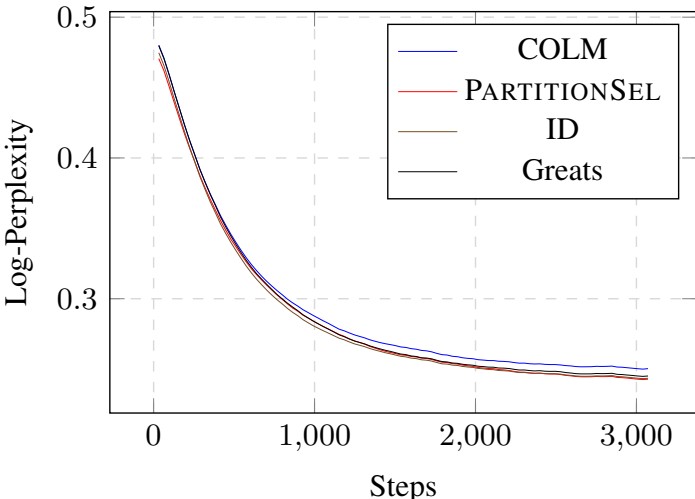

*Figure 8.* Validation log-pplx on MetaMathQA of LLama3b

### G.2. Llama3.2-8B

Table **??** reports the performance of LLAMA3.2-3B on MetamathQA (Yu et al., 2023). We train the model for 2048 steps, with $\hat{S}_t$ as 128 and $\kappa$ as 16, thus admitting a subset selection ratio of 12.5%

| Method | NumGLUE | MMLU-Math | GSM8K | SVAMP | SimulEq | DeepMind | AQuA | SAT | Avg |
|---|---|---|---|---|---|---|---|---|---|
| COLM | 38.00 | 40.35 | 48.22 | 60.80 | 17.12 | 16.60 | 31.50 | 39.09 | 36.46 |
| GradNorm | 36.85 | 39.53 | 44.81 | 54.10 | 11.28 | 15.80 | 31.50 | 40.91 | 34.35 |
| Greats-Independent | 41.27 | 36.65 | 48.90 | 58.40 | 18.87 | 15.80 | 34.25 | 38.64 | 36.60 |
| PARTITIONSEL | 43.95 | 39.01 | 49.13 | 56.70 | 19.65 | 15.90 | 36.61 | 44.09 | 38.13 |

*Table 6.* Accuracy (%) on evaluation datasets for Llama-3.2-3B trained on MetaMathQA with a 12.5% subset ratio.

| Method | NumGLUE | MMLU-Math | GSM8K | SVAMP | SimulEq | DeepMind | AQuA | SAT | Avg |
|---|---|---|---|---|---|---|---|---|---|
| COLM | 47.02 | 37.68 | 58.07 | 71.70 | 24.51 | 23.30 | 46.06 | 46.36 | 44.34 |
| GradNorm | 47.60 | 45.99 | 59.21 | 71.00 | 22.96 | 22.60 | 40.55 | 45.91 | 44.48 |
| Greats-Full | 49.81 | 32.65 | 60.65 | 67.90 | 25.88 | 20.60 | 41.34 | 52.27 | 43.89 |
| PARTITIONSEL | 50.29 | 39.63 | 59.82 | 69.90 | 30.16 | 24.00 | 37.40 | 51.36 | 45.32 |

*Table 7.* Accuracy (%) on evaluation datasets for Llama-3.1-8B trained on MetaMathQA with a 12.5% subset ratio.

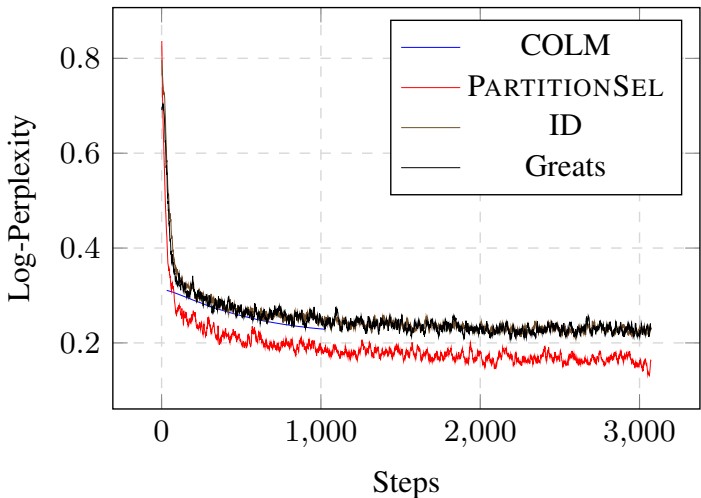

*Figure 9.* Validation log-pplx on MetaMathQA of LLama8b

### G.3. Qwem2.5-8B

Table **??** reports the performance of QWEN2.5-8B on MetamathQA (Yu et al., 2023). We train the model for 2048 steps, with $\hat{S}_t$ as 128 and $\kappa$ as 16, thus admitting a subset selection ratio of 12.5%

## H. Code

Our Code is available https://anonymous.4open.science/r/icmljoint-E292/

| Method | NumGLUE | MMLU-Math | GSM8K | SVAMP | SimulEq | DeepMind | AQuA | SAT | Avg |
|--------|---------|-----------|-------|-------|---------|----------|------|-----|-----|
| Greats-Independent | 57.68 | 64.68 | 80.52 | 84.60 | 41.44 | 37.90 | 60.24 | 78.18 | 63.15 |
| PARTITIONSEL | 56.43 | 64.58 | 83.47 | 86.40 | 38.72 | 38.10 | 63.78 | 83.64 | 64.39 |

*Table 8.* Accuracy (%) on evaluation datasets for Qwen2.5-7B trained on MetaMathQA with a 12.5% subset ratio.