# OpenReview forum: "Minibatch selection for Language Models via Partition Matroid Constrained Gradient Matching"
_ICML.cc/2026/Conference — ICML 2026 regular_

### Official Review · Reviewer_hYYh · 2026-02-26

**Soundness:** 2
**Presentation:** 2
**Significance:** 3
**Originality:** 2
**Overall Recommendation:** 3
**Confidence:** 3

**Summary:**

This paper focuses on how to jointly select samples across domains in each batch.  Specifically, the selection is under partition matroid constraint and this objective is theoretically weakly submodular.

**Compliance With Llm Reviewing Policy:**

Affirmed.

**Key Questions For Authors:**

1 There are several steps in the proposed pipeline that looks like computational expensive. It would be better to show the computational cost and memory cost in practical.

2 In section 6, the authors use Uniform Constraints at each iteration, which makes the per-domain budget proportional to the domain ratio. How robust is the proposed method when the domains are extremely imbalanced?

**Limitations:**

Due to the computational overhead and the fact that practical implementations may rely on many approximations that different from the theoretical objective presented in the paper, it remains debatable whether the method can be directly applied in real-world.

**Strengths And Weaknesses:**

Strengths:

1 This paper has a certain theoretical foundation.

2 The research problem studied of this paper is meaningful, how to jointly select samples from every batch.

Weaknesses

1 Figure 1 present the core motivation of this method, samples on the boundary vetween two domains have higher gradient similarity. The core motivation only illustrated via schematic figure.  in practice, it is unclear whrther this phenomenon is prevalent and severe enough to materially affect training. The authors do not give us detailed analysis or experimental results.

2 “our approach reduces the number of conflicting training gradient pairs significantly”, appear to be a mahor claimed benefit of the method, but I could not find any corresponding experimental results in the main paper.

3 The paper should be revised more carefully, because it contains many typos and unlear reference, such as :

       the caption of Figure 1, “… redundancy” should be “… redundancy.)”

       Line 154: “a an instance” should be “an instance”

       In abstract: “our analysis show” should be “our analysis shows”.

Minor: equation 12 is too wide.

---

> ### Author Rebuttal · Authors · 2026-03-31
>
> We thank the reviewer for their time and for the encouraging feedback on our submission. We address the questions raised by the reviewer below.
> > **Q1. Experimental evidence of phenomenon descibed in Figure 1**
>
> **A1.** Domain-wise independent selection only accounts for intra-domain gradient similarity, ignoring inter-domain interactions. In contrast, PartitionSel jointly considers both, reducing samples with redundant and conflicting gradients, an effect illustrated in Figure 1. Empirically, Figure 8 (Appendix G.1) confirms this: PartitionSel yields fewer conflicting gradient pairs ($\langle g_i, g_j \rangle < 0$) across training compared to independent selection. From Figure 8, we observe that:
> - Independent domain selection, which ignores cross-domain gradient interactions, consistently produces a higher number of conflicting gradient pairs throughout the training process.
> - PartitionSel's joint selection across domains substantially and consistently reduces conflicting gradient pairs throughout the training process.
>
> > **Q2. Experiments related to reduction in conflicting gradients pairs.**
>
> **A2.** As discussed in the previous Q1/A1, we present the conflicting gradient pairs analysis in Figure 8 (Appendix G.1). We missed referencing this figure in the main paper. In the camera ready version, we will discuss Figure 8 in the main paper.
>
> > **Q3. Regarding typos**
>
> **A3.** We thank the reviewer for the careful proofreading. We will correct the typos in the camera ready version.
>
> > **Q4. On the computational cost and memory overhead**
>
> **A4.** Figure 5 provides a detailed per-step latency breakdown for the proposed PartitionSel under 1-layer and 3-layer gradient regimes:
> - 1-layer: 2682ms total (2272ms gradient computation + 311ms subset selection)
> - 3-layer: 2822ms total (2468ms gradient computation + 315ms subset selection)
>
> We observe that the proposed OMP-based subset selection in PartitionSel accounts for only 11.6% of total step time in the 1-layer configuration, and this fraction decreases as more layers are used (~11.1%). Gradient computation dominates the computational cost — and this is common with all gradient-matching baselines ((Wang et al., 2024), (Nguyen et al., 2025)). The marginal overhead of PartitionSel's joint matroid-constrained OMP over a simple per-domain greedy scan is therefore small.
>
> We also discuss the utility of employing FJLT (Fast Johnson-Lindenstrauss Transform) compression to address memory costs in Table 4. Without FJLT compression, the naive per-example gradient computation causes OOM (out-of-memory) with 3 or more layers. With FJLT, PartitionSel scales to 5 layers without OOM on Qwen2.5-3B with LoRA rank 64. This demonstrates that the FJLT projection is essential and effective for scalability.
>
> For further details on end-to-end efficiency and memory details we guide the reviewer to Reviewer gvy7's A1/Q1 for reference.
>
> > **Q5: Robustness to Extreme Domain Imbalance**
>
>
>
>
> **A5.**  We compare the weighting scheme (line 311) for data mixture experiments and compare against DoReMI. Additional results are in [Figure 2](https://anonymous.4open.science/r/rebuttal-D25B/) where on the minority domains our method shows better eval log perplexity than DoReMI.*
>
> We note that the MetaMathQA dataset used in our experiments is already imbalanced. As discussed in Section 8.1 of the main paper and in Figure 7 (Appendix), MetaMathQA consists of eight domains with 0.19 imbalance ratio (minority/majority domain counts) (please also refer to table in Reviewer gvy7 Q2/A2 for domain counts). Our proposed approach PartitionSel consistently outperforms all the baselines on this (imbalanced) benchmark as shown in Tables 1,2,7,8 demonstrating practical robustness of our approach.
>
>
> Finally, we also note that the weighting scheme employed in our experiments (line 311) is a just a convenient default scheme. It can be modified based on prior knowledge and it only effects the parameters of the partition matroid. Thus, our Algorithms 1 and 2 remain unchanged and our overall pipeline is compatible with alternative weighting schemes as well.
>
> > **Q6: On "practical implementations may rely on many approximations that different from the theoretical objective presented in the paper"**
>
> **A6.** The different approximations employed in our approach are:
> - the first-order Taylor approximation of the validation loss (standard in previous methods (Wang et al., 2024, Killamsetty et al., 2021b)
> - Hessian approximation due to high computational overheads of computing the Hessian (as in existing works e.g. Wang et al., 2024, [1,2])
> - The proposed greedy OMP algorithm solves the proposed weakly submodular formulation with partition matroid constraints (but with a theoretical guarantee as discussed in Section 5).
>
> [1] Optimizing Neural Networks with Kronecker-factored Approximate Curvature, Martens et al
>
> [2] Model-Agnostic Meta-Learning for Fast Adaptation of Deep Networks, Finn et al.

---

> > ### Author Rebuttal · Reviewer_hYYh · 2026-04-05
> >
> > Thanks for your repsonse, I will raise my score.

---

> > > ### Author Response · Authors · 2026-04-06
> > >
> > > Dear Reviewer hYYh,
> > >
> > > We thank you for your positive feedback. In our portal, your raised score is yet to be reflected. It would be great if you could look into this issue.
> > >
> > > Regards,
> > >
> > > Authors

---

### Official Review · Reviewer_J9Lv · 2026-03-02

**Soundness:** 3
**Presentation:** 3
**Significance:** 3
**Originality:** 3
**Overall Recommendation:** 4
**Confidence:** 3

**Summary:**

This paper proposes a joint sample selection framework for training large language models that optimizes domain coverage and diversity using partition matroid constraints and gradient matching techniques.  It proposes a joint, domain-aware mini-batch selection method based on partition matroid constraints and formulates the selection as a utility maximization problem guided by validation loss reduction. It proposes an algorithm PARTITIONSEL that employs a matroid-based subset selection and uses Orthogonal Matching Pursuit (OMP) for efficient weakly submodular maximization under domain constraints.  Theoretical results guarantee the quality of selected subsets, with approximation ratios depending on RSC and RSM constants.  Experiments shows its performance improvements over several existing methods.

**Compliance With Llm Reviewing Policy:**

Affirmed.

**Final Justification:**

After reading the rebuttal and the rebuttal discussions of other reviews, I think the authors presented convincing arguments for their motivation, method design and explanation of experimental results.   Thus, I would change my final evaluation from 3 (weak reject) to 4 (weak accept).   Some concerns raised in my review will still need to be discussed in the final version, but the quality of the paper is good and should be accepted based on my view.

**Key Questions For Authors:**

As explained in the weakness part above, we would like to see the following answer on the weakness.

1.	The method PARTITIONSEL exploits domain-aware mini-batch selection method based on partition matroid constraints. No semantics nor embedding properties were explored.  Some discussion could be needed to see whether semantics or embedding properties could be explored to improve the performance.

2.	The improvement, as shown in Table 1, seems to be small and inclusive since the performance on a good set of data may not demonstrate the best performance although the average is 0.6363 vs. 0.6263 over ID.   Will this be sufficient to demonstrate the best performance?

3.	The overview of related work seems to be too brief and only along one line of work.

4.	There are many scattered grammar mistakes.   Table numbers are almost all dangling in the Appendix.

**Limitations:**

There is no discussion on the limitations and potential negative societal impact of their work.   Some discussions on cost, efficiency and scalability and other issues should be included in the discussion.

**Strengths And Weaknesses:**

Strength
1.	This paper introduces a framework for selecting training samples across multiple domains to improve large language model training efficiency and performance. It proposes a joint, domain-aware mini-batch selection method based on partition matroid constraints and formulates the selection as a utility maximization problem guided by validation loss reduction.
2.	It uses a weakly submodular objective function, enabling efficient orthogonal matching algorithms with theoretical guarantees and demonstrates significant empirical improvements over independent domain selection baselines on mathematical reasoning and molecular generation benchmarks. It also shows that the approach reduces gradient conflicts and redundancy across domains, leading to more diverse and effective training subsets.
3.	It introduces a validation-guided utility function balancing relevance and diversity, utilizes a weakly submodular objective, allowing approximation guarantees via greedy algorithms and establishes the monotonicity and weak sub-modularity of the proposed utility function.
4.	It proposes an algorithm PARTITIONSEL that employs a matroid-based subset selection and uses Orthogonal Matching Pursuit (OMP) for efficient weakly submodular maximization under domain constraints.  Theoretical results guarantee the quality of selected subsets, with approximation ratios depending on RSC and RSM constants.
5.	The performance study shows some improvements over some existing methods.

Weakness
1.	The method PARTITIONSEL exploits domain-aware mini-batch selection method based on partition matroid constraints. No semantics nor embedding properties were explored.  Some discussion could be needed to see whether semantics or embedding properties could be explored to improve the performance
2.	The improvement, as shown in Table 1, seems to be small and inclusive since the performance on a good set of data may not demonstrate the best performance although the average is 0.6363 vs. 0.6263 over ID.   Will this be sufficient to demonstrate the best performance?
3.	The overview of related work seems to be too brief and only along one line of work.
4.	There are many scattered grammar mistakes.   Table numbers are almost all dangling in the Appendix.

---

> ### Author Rebuttal · Authors · 2026-03-31
>
> We thank the reviewer for their time and for the encouraging feedback on our submission. We hereby address the questions raised by the reviewer.
>
> -----
>
> > **Q1. On semantics and embedding properties**
>
> **A1.** While our method does not use static semantic embeddings (e.g., from a frozen sentence encoder), it inherently relies on a rich, dynamic representation of the data. As detailed in Section 7 of the paper, our PartitionSel operates on the last-layer LoRA gradients. In the context of LLMs, these gradients effectively act as task-aware, dynamic embeddings. Unlike static semantic embeddings that only capture textual similarity, gradients capture how the model interprets the semantics relative to the current task and its current optimization state. Therefore, by matching validation gradients and minimizing gradient conflicts, PartitionSel is implicitly performing a highly optimized form of semantic selection that prioritizes samples the model currently finds most informative. To reinforce this, works like (Zhao et al[1], Wang et al., 2024, Nguyen et al, 2024) which are fully based on gradient matching showcase better performance than static or sentence-BERT embeddings (SBERT).
>
> However, we agree with the reviewer that exploring static semantic embeddings could yield further improvements in specific settings. Investigating those settings could be interesting directions of future work.
>
> > **Q2. Regarding "The improvement, as shown in Table 1, seems to be small ..."**
> > **A2.** We note in Table 1 that the average gain of our approach PartitionSel over eight tasks is *+1.00% over the best baseline ID* (domain wise indepedent selection) (0.6363 vs ID's 0.6263).
>
> The average, however, obscures substantial gains on specific datasets. For instance, the proposed PartitionSel achieves a +3.63% improvement on SAT task (0.7545 vs. ID's 0.7182) and a +3.11% improvement on SimulEq task (0.5467 vs. COLM/GradNorm's 0.5156). These are significant margins on complex tasks, demonstrating that our method effectively handles complex multi-step reasoning where existing baselines struggle. In addition, unlike methods that overfit to specific distributions (e.g., GREATS drops significantly on SimulEq, while ID drops on AQuA), the proposed PartitionSel maintains highly robust performance across the tasks, without catastrophic degradation on any single task.
>
> **The gains of PartitionSel generalize across models.** While Table 1 shows results for Qwen2.5-3B, Tables 6, 7, and 8 in the Appendix extend this to:
> - Llama-3.2-3B (+1.53 avg. over best baseline),
> - Llama-3.1-8B (+0.98 avg. over best baseline),
> - Qwen2.5-7B (+1.24 avg. over best baseline).
>
> Consistent improvements across models spanning 3B to 8B parameters are strong evidence of our approach's performance. Furthermore, the log perplexity curves (Fig 2 in the main paper and Figure 10 in the Appendix) show that PartitionSel achieves consistent lower validation log pplx than the baselines.
>
> **The gains of PartitionSel generalize across tasks as well.**
>
> - **Molecular generation (Table 2):** On the molecular generation benchmark (Mol-Instructions), PartitionSel achieves the best BLEU or Edit-distance score in 6 out of 8 evaluation settings, with an average BLEU gain of +2.6 (normalised by 100 in Table 2) and an average Edit-distance improvement of 1.86 over the best baseline.
>
> - **Comparison with DoReMi (Table 3):** PartitionSel outperforms DoReMi, a significantly more expensive method requiring proxy model training, by +4.65 average accuracy points across MMLU-Math, AQuA, and SAT.
>
> We note that all baselines and PartitionSel are trained on the same data (MetaMathQA) under the same conditions. The results therefore directly measure the contribution of PartitionSel's selection mechanism.
>
> > **Q3. The overview of related work seems to be too brief**
>
> **A3.** We thank the reviewer for the suggestion. In the camera ready version, we will include additional papers apart from the ones already discussed (in introduction and related works sections) with more details.
>
> > **Q4. Grammar mistakes**
>
> **A4.** We thank the reviewer for the careful proofreading. We will correct the typos in the camera ready version.
>
> > **Q5. On the limitations and potential negative societal impact**
>
> **A5.** As suggested by the reviewer, we will include a discussion on the limitations and potential negative societal impact in the camera ready version. Please also refer to Reviewer gvy7 Q1/A1 and Reviewer hYYh Q5/A5 for a detailed discussion of the end-to-end efficiency of the proposed approach.
>
>
> [1] Beyond Similarity: A Gradient-based Graph Method for Instruction Tuning Data Selection, Zhao et al, 2025

---

> > ### Author Rebuttal · Reviewer_J9Lv · 2026-04-04
> >
> > I have read the rebuttal from authors, and some major concern of the work has been resolved.  I will read it through to see if my overall score needs to be adjusted.

---

### Official Review · Reviewer_JLhq · 2026-03-11

**Soundness:** 3
**Presentation:** 3
**Significance:** 3
**Originality:** 3
**Overall Recommendation:** 4
**Confidence:** 3

**Summary:**

Domain mixing is a popular technique to take advantage of data in a more complete, grounded way. However, many existing methods either rely on very expensive techniques to compute domain mixture weights or are not theoretically grounded. This work shifts from learning weights on each class to instead deciding which samples to include per batch. The proposed method, coming from a first-order Taylor expansion of the loss around a gradient update, happens to be weakly submodular, allowing for efficient techniques from convex analysis to ensure a high level of performance from a greedy algorithm. This algorithm is compared against a handful of baselines, showing superior performance on multiple models across multiple scales.

**Compliance With Llm Reviewing Policy:**

Affirmed.

**Final Justification:**

Batch-wise sample selection offers a complementary perspective to existing approaches that operate at the dataset level. The method appears sound, and the paper is clearly written. My main concern regarding the limited literature review has been addressed, which is particularly valuable given the field’s emphasis on data cleaning and domain mixing, both of which have closely related goals.

These connections also suggest that batch-wise sample selection could motivate new approaches within these adjacent areas. Overall, this work introduces a simple and lightweight method that advances data selection in a novel and underexplored direction.

**Key Questions For Authors:**

1. Can a broader literature survey be conducted and included, framing this work and its novelty compared to other cluster weighting methods? The idea looks novel, but to readers not familiar with similar works, it would be hard to understand where this novelty is.
2. What are the columns in Figure 3? PartitionSel is red in Figure 2 but yellow in Figure 4.
3. Can the best result for each method in Table 1 be bolded, and the second best underlined? Same for Table 2. It's a little hard to parse for the best method in each case.
4. The example in section 8.2 is a little confusing. Does this mean the models being fine-tuned are designed proteins in the SMILES format? Designing molecules makes sense for that granularity of a format, but a protein seems prohibitively complex to solve, even with frontier models.

Minor Typos:
- Line 218, heterogeneour -> heterogenous
- Line 182, == -> =
- Line 199, domain wise -> domain-wise
- Line 1097, broken ref

**Limitations:**

yes

**Strengths And Weaknesses:**

Strengths
- The method and the mathematical justification of such appear sound.
- The paper flowed nicely, integrating a lot of convex analysis cleanly into the writing. Other related works most relevant to this one have their methods clearly described, also seamlessly part of the writing.
- Such a per-batch, light-weight sub-sampling method is quite useful. Other works use domain alignment as a principle for sampling, although this is usually done at a class level rather than at a sample level. Integrating a more nuanced, lightweight approach that would be better suited to remove outliers and such is quite appealing.
- The experiments cover a variety of datasets and models

Weaknesses
- The related work section is quite short. Domain mixing is a large field with a wide number of studies looking at different methods to compute component weights. Only having 6 papers mentioned is a bit light, especially since there is room within the page limit. A slightly larger literature review, possibly added in the appendix if brevity is desired in the main body, would really help tie together this paper with other existing works.
- A few pieces of the writing and the results are a little unclear. Some are listed in the questions below.

---

> ### Author Rebuttal · Authors · 2026-03-31
>
> We thank the reviewer for their time and for the encouraging feedback on our submission. We appreciate that the reviewer found the proposed method to be lightweight and its underlying theory to be sound and relevant. We hereby address the questions raised by the reviewer
>
> > **Q1. Can a broader literature survey be conducted and included, framing this work and its novelty compared to other cluster weighting methods? The idea looks novel, but to readers not familiar with similar works, it would be hard to understand where this novelty is.**
>
> **A1.** We thank the reviewer for the suggestion. In the camera ready version, we will include additional papers apart from the ones already discussed (in introduction and related works sections) with more details.
>
> > **Q2. What are the columns in Figure 3? PartitionSel is red in Figure 2 but yellow in Figure 4.**
>
> **A2.** In Figure 3, the red column denotes our proposed PartitionSel approach while the blue column denotes the Random selection baseline. We thank the reviewer for highlighting the color scheme inconsistency for the methods. We will standardize the legends across all figures in the revision.
>
> > **Q3. Can the best result for each method in Tables 1 and 2 be bolded, and the second best underlined?**
>
> **A3.** Yes, we will bold the best result and underline the second-best results in all the tables to improve readability of the results.
>
> > **Q4. In Section 8.2, Does this mean the models being fine-tuned are designed proteins in the SMILES format?**
>
> **A4.** The SMILES format is only required for molecule generation experiments and not for protein design experiments. While the Mol-instructions dataset (Fang et al., 2023) contains three components: molecule-oriented instructions, protein-oriented instructions, and biomolecular text instructions, we employed the SMILES format only to molecule-oriented instructions in our experiments. We will clarify in Section 8.2 in the camera ready version.
>
> > **Q5. Regarding minor typos**
>
> **A5.** We thank the reviewer for the careful proofreading. We will correct the typos in the camera ready version.

---

> > ### Author Rebuttal · Reviewer_JLhq · 2026-04-01
> >
> > Thank you to the authors for the responses. With a larger literature review, the main concern of mine has been addressed. Additionally, correcting the typos and writing improves the overall quality of the paper. Based on these updates, I will adjust my score from a 3 to a 4.

---

### Official Review · Reviewer_gvy7 · 2026-03-12

**Soundness:** 2
**Presentation:** 3
**Significance:** 3
**Originality:** 2
**Overall Recommendation:** 4
**Confidence:** 3

**Summary:**

This paper studies minibatch subset selection for large language model fine-tuning in heterogeneous multi-domain settings. The authors formulate the selection objective as maximizing a validation-guided gradient matching utility subject to partition matroid constraints (enforcing per-domain budgets). Crucially, they resolve the non-monotonicity issue present in prior works (like GREATS) by introducing non-negative sample weights, proving that the resulting objective is monotonic and weakly submodular. An Orthogonal Matching Pursuit (OMP) algorithm is employed for selection. Extensive experiments on mathematical reasoning (Qwen2.5, Llama-3) and molecular generation demonstrate consistent improvements over independent domain selection and domain-agnostic baselines.

**Compliance With Llm Reviewing Policy:**

Affirmed.

**Key Questions For Authors:**

1. Could you provide a comprehensive comparison of the end-to-end wall-clock training time and peak GPU memory usage for PARTITIONSEL versus GREATS, COLM, and the Random baseline?
2. In Section 6, the domain budget is set proportionally. How does PARTITIONSEL perform if the dataset contains severe class imbalances where minority domains are highly relevant to the validation set? Does the strict proportional budget hinder performance compared to dynamically learned mixtures like DoReMi?
3. The empirical gains are clearly demonstrated for SFT. Are there any theoretical or computational roadblocks to applying this partition matroid gradient matching framework to LLM continuous pre-training?

**Limitations:**

No. The paper should discuss limitations more explicitly, especially regarding computational overhead, sensitivity to domain budget design, and the current scope of empirical validation.

**Strengths And Weaknesses:**

## Strengths
1. **Strong Theoretical Grounding**: The formulation successfully bridges partition matroids with gradient matching. More importantly, the introduction of importance weights resolves the non-monotonicity flaw in the GREATS baseline, rendering the objective monotonic and weakly submodular, accompanied by solid theoretical guarantees (Lemma 5.1, Theorem 5.1).
2. **Clean Conceptual Formulation**: Representing multi-domain batch constraints as a partition matroid is elegant and bypasses the need for training expensive proxy models for continuous domain weighting (e.g., DoReMi).
3. **Solid Empirical Setup**: The authors evaluate across two distinct, complex tasks (mathematical reasoning and molecule generation) using modern models up to 8B parameters, proving the method's efficacy beyond a single niche task.

## Weaknesses
1. **End-to-End Efficiency Transparency**: While the paper discusses memory-efficient gradient computation via FJLT and provides a latency breakdown for a single step (Figure 5), the paper lacks a transparent comparison of the *end-to-end wall-clock training time*. Since computing validation gradients and solving the OMP at each step introduces overhead, the total cost versus the performance trade-off against a simple random-selection baseline needs clearer documentation.
2. **Handling Extreme Imbalance under Proportional Constraints**: The method dynamically assigns domain constraints (kappa) proportional to the domain's size in the candidate batch (Section 6). While this avoids manual tuning, it raises the question of whether this rigid proportional constraint artificially caps the sampling of a highly informative but minority domain in heavily skewed datasets.
3. **Scope of Application**: The empirical validation is thoroughly conducted in the Supervised Fine-Tuning (SFT) setting. It is unclear if the assumptions (e.g., access to a clean validation set that perfectly aligns with training goals) hold or scale effectively to the Pre-training phase, where data mixtures are significantly larger and more diverse.

---

> ### Author Rebuttal · Authors · 2026-03-31
>
> We thank the reviewer for their detailed assessment of our work and providing encouraging and thoughtful feedback. We appreciate that the reviewer acknowledged strong theoretical grounding, clean conceptual formulation, and solid empirical setup for our work. Below, we address the questions raised by the reviewer.
>
> ----
>
> > **Q1: While the paper discusses memory-efficient gradient computation via FJLT and provides a latency breakdown for a single step (Figure 5), the paper lacks a transparent comparison of the end-to-end wall-clock training time.**
>
> **A1.** As advised by the reviewer, we provide end-to-end wall clock time details for each baseline in the following table.
>
> Experimental setup: We use a batch size of 16 with a selection fraction k=0.25, training a Qwen2.5-3B model for 1K steps.
>
> | Method | Time (mins) |
> |--------|------------|
> | PARTITIONSEL (Ours) | 51:08 |
> | GREATS | 50.67 |
> | COLM | 47:29 |
> | ID | 53:25 |
> | Random | 44.17 |
>
> Overall, our method incurs comparable end-to-end training time relative to existing approaches, while achieving better accuracy and log-perplexity trade-offs as shown in our experiment analysis, demonstrating that its computational overhead is negligible in practice.
>
> We additionally report the peak GPU memory cost as requested by the reviewer.
>
> | Method | Memory |
> |--------|--------|
> | PartitionSel (Ours) | 31.5 GB |
> | ID | 32.9 GB |
> | GREATS | 32.8 GB |
> | COLM | 32.7 GB |
> | Random | 28.0 GB |
>
> We observe that the peak GPU memory consumption of PartitionSel is similar to existing gradient-based baselines. Random consumes least peak memory as it does not perform subset selection and incurs no  gradient computation.
>
> > **Q2: Handling extreme imbalance under proportional constraints**
>
> **A2.** In our current data mixture experiments, we employ MetaMathQA dataset for training, which is an imbalanced dataset. Figure 7 in Appendix (pg 16) shows the imbalance across different domains, where the minority domains are MATH_FOBAR and MATH_SV. In the following table, we provide the percentage of samples per domain (in MetaMathQA).
>
> | Domain | Percentage |
> |--------|------------|
> | GSM_AnsAug | 20.3% |
> | GSM_FOBAR | 10.1% |
> | GSM_Rephrased | 20.3% |
> | GSM_SV | 10.1% |
> | MATH_AnsAug | 18.9% |
> | MATH_Rephrased | 12.7% |
> | MATH_FOBAR | 3.8% |
> | MATH_SV | 3.8% |
>
> In Figure 4 we showcase the domain weights assigned by PartitionSel where the minority domain weights are within similar range as assigned by DoReMI, thereby showing that the domain weight scheme of PartitionSel (line 311) is robust to domain imbalance settings.
> We provide the log perplexity curves in external [Figure 2](https://anonymous.4open.science/r/rebuttal-D25B), where PartitionSel has better log-pplx reduction than DoReMi by 12.5% and 9.4% on minority(MATH_FOBAR, MATH_SV) domains, respectively, indicating it performs well in on imbalanced domains.
> We spell out a few values from the log-pplx plot in table form here:
> | Step | MATH_SV DoReMi | MATH_SV PartitionSel | MATH_FOBAR DoReMi | MATH_FOBAR PartitionSel |
> |-----:|---------------:|---------------------:|------------------:|------------------------:|
> | 128 | 0.40 | **0.35** | 0.34 | **0.32** |
> | 256 | 0.36 | **0.33** | 0.31 | **0.29** |
> | 512 | 0.35 | **0.30** | 0.29 | **0.28** |
> | 768 | 0.33 | **0.30** | 0.29 | **0.27** |
> | 1024 | 0.32 | **0.29** | 0.28 | **0.27** |
> | 1536 | 0.32 | **0.28** | 0.28 | **0.26** |
> | 2048 | 0.32 | **0.28** | 0.28 | **0.26** |
>
> > **Q3: It is unclear if the assumptions (e.g., access to a clean validation set ) hold or scale effectively to the Pre-training phase, where data mixtures are significantly larger and more diverse.**
>
> **A3.** While our primary empirical validation is in the supervised fine tuning setting, we follow (Wang et al., 2024) to implement pretraining as well. Since pretraining with DoReMI is a computational constraint for us as it requires additional overheads such as pretraining an additional proxy model before the main model pretraining even begins, we compare against mini batch coreset selection baseline ID (Wang et al., 2024).
>
> We train a 1B Llama3.2 model on 1.5B tokens from a multi-domain corpus (Dolma [1]) spanning heterogeneous sources (Common Crawl, Wikipedia, StackExchange, Reddit; ~2.9M chunks total). We used a batch like of 16k tokens (32 sequences of length 512), learning rate of 3e-4 with linear decay, warmup of 1000 steps and AdamW. In the external [Figure 1](https://anonymous.4open.science/r/rebuttal-D25B) we observe that our approach achieves better log pplx reduction than domain wise independent selection by 4.53%. We spell out a few values from the log-pplx plot in table form here:
> | Step  | PartitionSel | ID    |
> |------:|-------------:|------:|
> | 5K    | **8.72**     | 8.80  |
> | 10K   | **7.58**     | 7.77  |
> | 20K   | **7.43**     | 7.63  |
> | 35K   | **7.35**     | 7.58  |
> | 50K   | **7.32**     | 7.55  |
>
> [1] Dolma: an Open Corpus of Three Trillion Tokens for Language Model Pretraining Research

---

> > ### Author Rebuttal · Reviewer_gvy7 · 2026-04-06
> >
> > Thank you for the detailed rebuttal. My main concerns have been adequately addressed. In particular, the rebuttal now provides end-to-end wall-clock time and peak GPU memory comparisons, which clarifies the practical overhead of the method. The response on domain imbalance is also helpful: the authors provide additional evidence on minority domains and show that the method remains competitive in imbalanced settings. Finally, the added discussion and preliminary evidence for the pre-training setting meaningfully broaden the scope beyond SFT. While some of these additions should be clearly incorporated into the final version of the paper, the rebuttal sufficiently addresses the concerns raised in my original review.

---

> > > ### Author Response · Authors · 2026-04-07
> > >
> > > We thank the reviewer for their thoughtful follow-up and for noting that the key concerns have been addressed. We’re glad the additional experiments and discussion clarified both the practical and broader impact of our work. We will incorporate the above discussed improvements clearly in the final version.
> > >
> > > Given this, we would appreciate it if you could consider updating your score to reflect the revised assessment.

---

### Decision · Program_Chairs · 2026-04-30

**Decision:**

Accept (regular)

**Comment:**

The paper proposes PARTITIONSEL, a joint minibatch selection framework for multi-domain LLM training that uses partition matroid constraints to maximize validation-guided gradient matching. By incorporating importance weights, the authors resolve non-monotonicity issues in prior work (e.g., GREATS), theoretically establishing the objective as monotonic and weakly submodular with provable Orthogonal Matching Pursuit (OMP) approximation guarantees.

Empirical evaluations across models  demonstrate consistent improvements in mathematical reasoning and molecular generation over competitive baselines. The rebuttal successfully addressed practical concerns regarding end-to-end efficiency, memory overhead, and domain imbalance, while showcasing scalability to pre-training; this led to a consensus for acceptance, recognizing the method as a principled alternative to expensive proxy-based weighting.